# Tau and spectraplakins promote synapse formation and maintenance through Jun kinase and neuronal trafficking

Andre Voelzmann[1], Pilar Okenve-Ramos[2], Yue Qu[1], Monika Chojnowska-Monga[2], Manuela del Caño-Espinel[3], Andreas Prokop[1]*, Natalia Sanchez-Soriano[2]*

[1]Faculty of Life Sciences, The University of Manchester, Manchester, United Kingdom; [2]Department of Cellular and Molecular Physiology, Institute of Translational Medicine, University of Liverpool, Liverpool, United Kingdom; [3]Instituto de Biología y Genética Molecular-Departamento de Bioquímica y Biología Molecular y Fisiología, Universidad de Valladolid-CSIC, Valladolid, Spain

**Abstract** The mechanisms regulating synapse numbers during development and ageing are essential for normal brain function and closely linked to brain disorders including dementias. Using *Drosophila*, we demonstrate roles of the microtubule-associated protein Tau in regulating synapse numbers, thus unravelling an important cellular requirement of normal Tau. In this context, we find that Tau displays a strong functional overlap with microtubule-binding spectraplakins, establishing new links between two different neurodegenerative factors. Tau and the spectraplakin Short Stop act upstream of a three-step regulatory cascade ensuring adequate delivery of synaptic proteins. This cascade involves microtubule stability as the initial trigger, JNK signalling as the central mediator, and kinesin-3 mediated axonal transport as the key effector. This cascade acts during development (synapse formation) and ageing (synapse maintenance) alike. Therefore, our findings suggest novel explanations for intellectual disability in Tau deficient individuals, as well as early synapse loss in dementias including Alzheimer's disease.

*For correspondence: Andreas. Prokop@manchester.ac.uk (AP); N.Sanchez-Soriano@liverpool.ac. uk (NS-S)

**Competing interests:** The authors declare that no competing interests exist.

## Introduction

The correct formation and subsequent maintenance of synapses is a key prerequisite for brain development, function and longevity. Precocious loss of synapses is observed in late onset neurodegenerative diseases including Alzheimer's disease (AD) and Frontotemporal Dementia (FTD), likely contributing to the cognitive decline and neuronal decay observed in patients (*Pooler et al., 2014*; *Saxena and Caroni, 2007*; *Serrano-Pozo et al., 2011*). Therefore, the characterisation of mechanisms maintaining synapses during ageing would have major implications for our understanding of dementias.

The development of synapses and their maintenance during ageing is dependent on sustained transport of synaptic proteins from the distant soma, driven by motor proteins which trail along the bundles of microtubules in axons and dendrites (*Goldstein et al., 2008*). Microtubules are regulated by microtubule binding proteins which are therefore in a key position to regulate synapse formation and maintenance (*Prokop, 2013*).

Tau is a microtubule associated protein (MAP) discovered in the mid-seventies (*Weingarten et al., 1975*). Reduction in Tau levels has been linked to intellectual disability (*Sapir et al., 2012*) and a class of brain disorders termed 'dementias which lack distinctive histopathology' (DLDH) (*Zhukareva et al., 2001*). Tau detachment from MTs is linked to prominent neurodegenerative diseases such as Alzheimer's disease, Frontotemporal Dementia and some forms of

**eLife digest** Nerve cells form cable-like projections called axons that connect to other nerve cells to form the nervous system. Axons carry nerve impulses in the form of electrical messages, and they pass on these messages to other cells at junctions known as synapses. Specific patterns of connections between axons allow us to coordinate our movements, feel emotions and think. In Alzheimer's disease and other neurodegenerative conditions, synapses often decay earlier than they should, which can cause important connections between nerve cells to be lost.

To be able to make and maintain synapses, nerve cells transport materials from the main body of the cell along axons to the sites where synapses form. A protein called Tau and a family of proteins called the spectraplakins are linked to neurodegenerative diseases. Changes (or mutations) in these proteins were known to disrupt the formation and maintenance of synapses, but it was not clear how these proteins work in this context.

Voelzmann et al. studied Tau and spectraplakin during synapse formation and maintenance in fruit flies. The experiments show that both proteins stabilise tube-like structures called microtubules in axons, which provide structural support to cells. The loss of Tau or spectraplakins causes the microtubules to fall apart and triggers an internal stress signalling pathway known as the JNK pathway. Activating JNK signalling blocks the transport of synaptic materials along axons, which prevents the formation of new synapses and starves existing synapses leading to their decay.

The next step is to find out whether Tau and spectraplakins play similar roles in the nerve cells of mammals, which may open up new opportunities to develop therapies for Alzheimer's and other neurodegenerative diseases.

Parkinson's disease (*Kovacs, 2015*). In vitro, Tau has the ability to regulate microtubule properties including stability, cross-linkage and polymerisation (*Morris et al., 2013*). Through such functions, Tau would be expected to regulate multiple aspects of neuronal cell biology, but its physiological roles are still not understood and highly debated (*Morris et al., 2013*). This might partly be due to experimental challenges posed by functional redundancy, where other MAPs are proposed to mask physiological roles of Tau (*Ma et al., 2014*; *Takei et al., 2000*).

A good model in which to deal with functional redundancy is the fruit fly *Drosophila melanogaster*. As is ideal for studies of Tau, *Drosophila* neurons provide access to powerful genetics, they are readily established for research on the neuronal cytoskeleton (*Sánchez-Soriano et al., 2010*), on neuronal transport (*Schwarz, 2013*) and on synapses (*Prokop and Meinertzhagen, 2006*). Importantly, concepts and mechanisms gained from work in flies are often well conserved in higher organisms (*Bellen et al., 2010*; *Jaiswal et al., 2012*).

Work in *Drosophila* suggested that the spectraplakin Short Stop (Shot), a large actin-MT linker molecules and potent regulators of microtubules, could display potential functional overlap with Tau during microtubule stabilisation (*Alves-Silva et al., 2012*; *Prokop, 2013*). This hypothesis is attractive because the well-conserved mammalian spectraplakin Dystonin is already linked to a neurodegenerative disease (type VI hereditary sensory autonomic neuropathy; OMIM #614653;) (*Ferrier et al., 2013*), and its paralogue ACF7/MACF1 plays important roles during brain development (*Goryunov et al., 2010*; *Ka and Kim, 2015*). Since ACF7 continues to be expressed in the brain, it is tempting to speculate that it might be required for neuronal maintenance (*Bernier et al., 2000*).

Here we use *Drosophila* neurons, in culture and in vivo alike, to demonstrate novel roles of Tau in regulating the formation and maintenance of synapses during ageing, by coordinating the intracellular trafficking of synaptic proteins. Thus, we show that the role of Tau in synapse regulation occurs in functional overlap with Shot. The robust *shot-tau* double-mutant phenotypes enabled us to study the mechanistic cascade composed of three steps: microtubule stability as the trigger, the JNK signalling pathway as the mediator and kinesin-3 mediated axonal transport of synaptic proteins as the key effector. We propose a new mechanism based on the loss of Tau function which could explain intellectual disability in MAPT (the human tau gene) mutant individuals and precocious synapse loss in tau-related neurodegeneration (*Saxena and Caroni, 2007*; *Serrano-Pozo et al., 2011*).

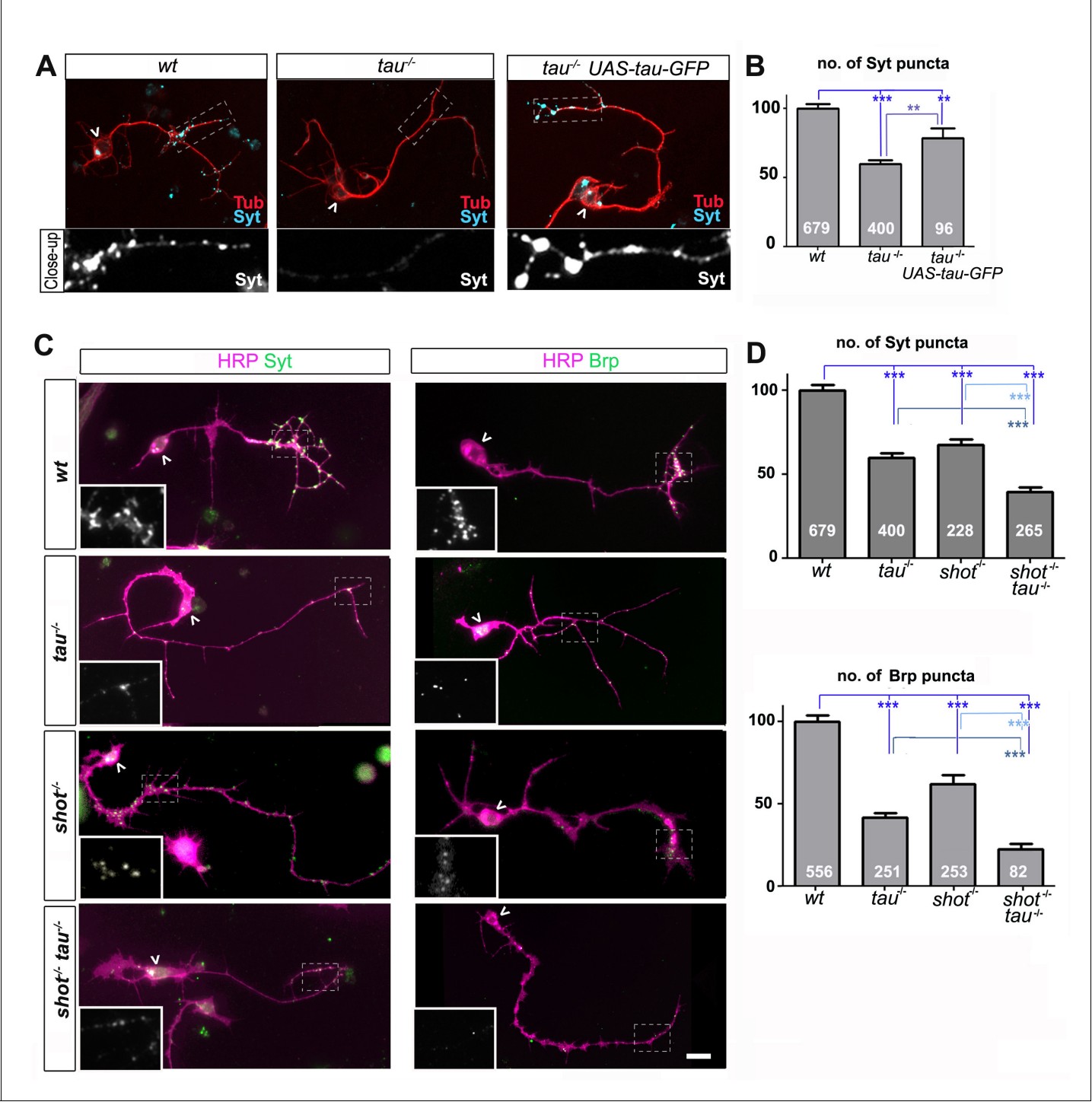

**Figure 1.** Tau and Shot are required for the formation of synaptic specialisations in axons. (**A**) Primary neurons at 2 DIV obtained from embryos that were wildtype (wt), *tau*[-/-], and *tau*[-/-] with *elav-Gal4* driven expression of *UAS-tau-GFP*; neurons were stained for tubulin (Tub, red) and the synaptic protein Synaptotagmin (Syt, light blue). (**B**) Quantification of the experiment shown in A, shown as the number of Syt puncta per neuron, normalised to wildtype (the assessed numbers of neurons are indicated in each bar, ***$P_{MW}$<0.001, **$P_{MW}$<0.01). (**C**) Primary *Drosophila* neurons at 2DIV, obtained from embryos that were wildtype (wt), *tau*[MR22](*tau*[-/-]), *shot*[3](*shot*[-/-]), and *shot*[3] *tau*[MR22] (*shot*[-/-] *tau*[-/-]), co-stained with antibodies against HRP (magenta) and the synaptic proteins (green) Syt and Bruchpilot (Brp); areas emboxed with dashed lines are displayed as magnified insets showing the synaptic staining only. (**D**) Quantification of the experiments in C, displayed as number (no.) of Syt and Brp puncta per neuron, normalised to wildtype (the assessed numbers of neurons are indicated in each bar, ***$P_{MW}$<0.001; **$P_{MW}$<0.01; *$P_{MW}$<0.05,). Scale bar: 10 μm. A statistics summary of the data shown here is available in *Figure 1—source data 1*.

*Figure 1 continued on next page*

*Figure 1 continued*

The following source data and figure supplements are available for figure 1:

**Source data 1.** Summary of the statistics from *Figure 1B and C*.

**Figure supplement 1.** Co-localisation of presynaptic markers reveals presynaptic specialisations.

**Figure supplement 2.** Rescue experiments with Shot and Tau demonstrate redundant roles in synapses.

**Figure supplement 2—source data 1.** Summary of the statistics from *Figure 1—figure supplement 2B*.

## Results

### Tau is required for the formation of synapses

To study synaptic roles of *Drosophila* Tau, we first used primary *Drosophila* neurons generated from *tau* mutant embryos. Primary fly neurons are genetically and experimentally highly amenable and provide robust cellular and subcellular readouts (*Prokop et al., 2012*). These cultures are also particularly suited for the study of embryonic lethal mutations since they allow the examination of neurons beyond the embryonic lethal stage. Already 8 hr in vitro (HIV), these neurons show transport of synaptic material in the growing axon (*Sánchez-Soriano et al., 2010*) and after 2 days in vitro (DIV), they display functional presynaptic sites (*Küppers-Munther et al., 2004*; *Küppers et al., 2003*) that can be reliably stained with antibodies against presynaptic proteins (*Figure 1—figure supplement 1*). They contain dense bars and synaptic vesicle accumulations which undergo excitation-dependent uptake and release (*Küppers-Munther et al., 2004*; *Küppers et al., 2003*).

The *Df(3R)tauMR22* mutation (*tau^{MR22}*) is an embryonic lethal chromosome deletion that uncovers most of the *Drosophila tau* gene and is a true null allele (*Bolkan and Kretzschmar, 2014*; *Doerflinger et al., 2003*). We found that *tau^{MR22}* mutant primary neurons at 2 DIV show a decrease in the number of puncta positive for Bruchpilot (Brp) and Synaptotagmin (Syt) (Bruchpilot/Brp: 42%; Synaptotagmin/Syt: 59%; all compared to wildtype control neurons; *Figure 1*). Our finding suggest that tau-deficient primary neurons contain fewer Brp and Syt positive presynaptic specialisations. In the following, we will refer to this phenotype as synapse reduction.

To confirm that this reduction in synapse numbers was due to the loss of Tau, we performed rescue experiments using Gal4-induced neuronal expression of *UAS-tau-GFP* (*Doerflinger et al., 2003*) in *tau^{MR22}* mutant neurons. We found a significant improvement of the *tau^{MR22}* mutant phenotype (*Figure 1A–B*). We concluded that absence of Tau causes a reduction in presynaptic sites.

### Tau displays functional overlap with the spectraplakin Shot

To assess potential functional overlap of Tau with the *Drosophila* spectraplakin Shot, we first analysed *shot^3* mutant primary neurons at 2 DIV. We found a reduction in synapse numbers (Brp: 62%; Syt: 67%; *Figure 1C–D*), consistent with previous descriptions in vivo (*Löhr et al., 2002*; *Prokop et al., 1998*). We confirmed that this reduction in synapse numbers was due to the loss of Shot by using Gal4-induced neuronal expression of *UAS-shot-GFP* (*Alves-Silva et al., 2012*; *Sanchez-Soriano et al., 2009*) which significantly rescued the synapse phenotype in *shot^3* mutant neurons (*Figure 1—figure supplement 2*), confirming the involvement of Shot.

We then assessed potential functional overlap of Shot and Tau. First, we analysed primary neurons double-mutant for the *shot^3* and *tau^{MR22}* null alleles (*shot-tau*) which showed even lower synapse numbers (Brp: 22%; Syt: 39%; *Figure 1C–D*) than either of the single mutant neurons. Notably, these analyses were performed on clearly polarised neurons with well developed axons to exclude indirect effects caused by defective axon growth (*Figure 3—figure supplement 3*). Despite that, we found that the double-mutant neurons displayed reduced branch numbers (*Figure 3—figure supplement 3F*). However, we could demonstrate that the lower number in branches is not the cause for synapse reduction by using knock-down experiments as well as rescue experiments (explained in

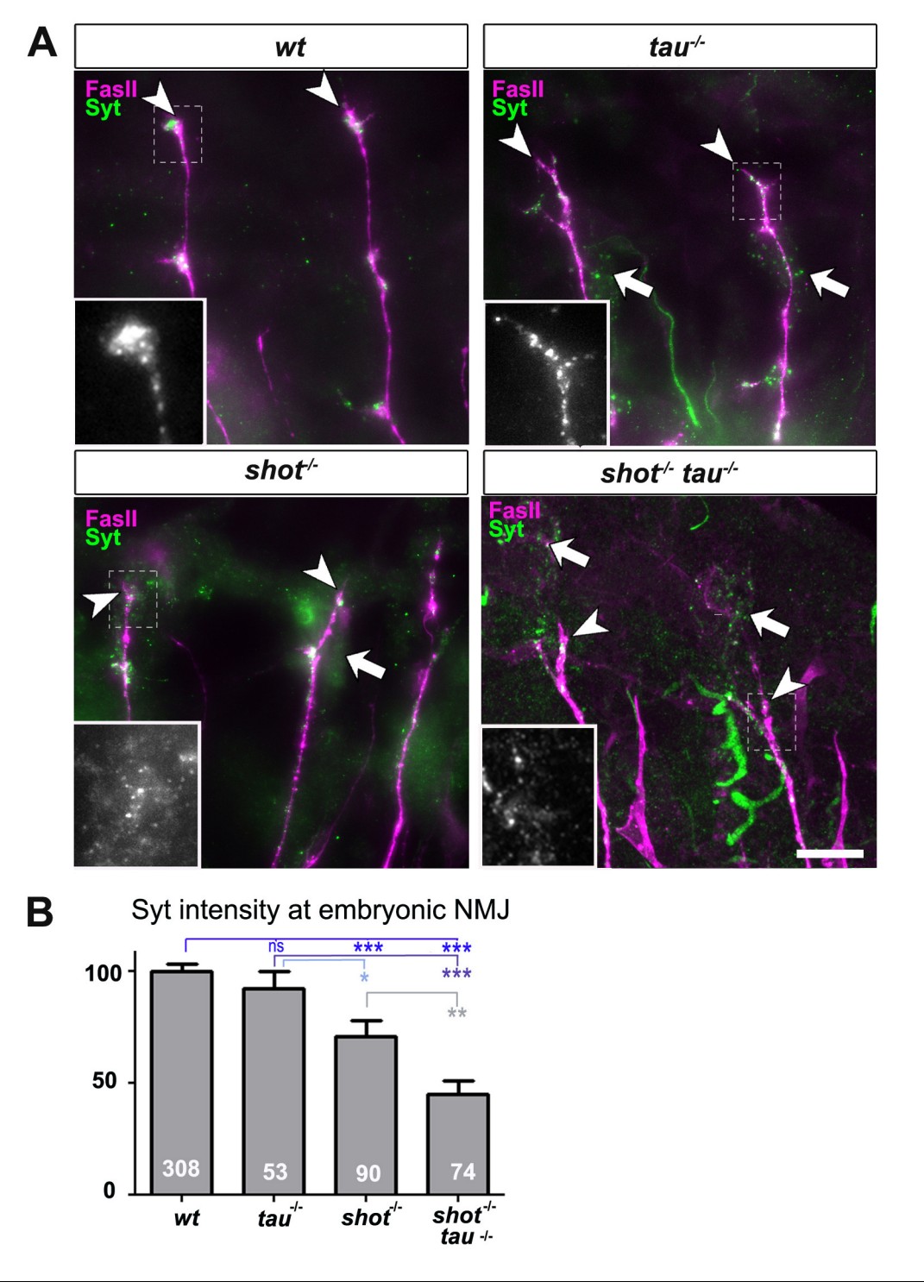

**Figure 2.** Tau and Shot regulate the localisation of presynaptic proteins at the embryonic NMJ in vivo. (**A**) Images show the dorsal segment of inter-segmental motornerves (*Landgraf et al., 2003*) in stage 16 embryos that were wildtype (wt), *tau^MR22* (tau^-/-), *shot^3* (shot^-/-), and *shot^3 tau^MR22* (shot^-/- tau^-/-), stained with antibodies against Syt (green) and the motorneuron-specific cell membrane protein Fasciclin II (FasII, magenta). Arrowheads depict the distal end of the motoraxons where the nascent NMJs are forming; boxed areas are displayed as enlarged insets showing anti-Syt staining only. Note that cell bodies of sensory neurons contain visible levels of Syt in the mutant (white arrows) but not in wildtype neurons (open arrow). (**B**) Quantification of the experiments in **A**, shown as the average intensity of Syt at the nerve ending normalised to wildtype (the sample number of NMJs is indicated in each bar, ***$P_{MW}$<0.001; **$P_{MW}$<0.01; *$P_{MW}$<0.05; ns, not significant $P_{MW}$>0.05). Scale bars: 10 µm. A statistics summary of the data shown here is available in *Figure 2—source data 1*.

*Figure 2 continued on next page*

*Figure 2 continued*

The following source data and figure supplement are available for figure 2:

**Source data 1.** Summary of the statistics from *Figure 2B*.

**Figure supplement 1.** Schematic drawings of embryonic tissues analysed in this study.

detail below, *Figure 3—figure supplement 3* and *Figure 3—figure supplement 4*, see also Discussion).

In further support of functional overlap, also our genetic interaction studies revealed a synapse reduction phenotype in *shot*$^{3/+}$ *tau*$^{MR22/+}$ double heterozygous mutant neurons (see later in Figure 5A). Finally, we performed cross-rescue experiments by expressing a *shot* transgene in *tau*$^{MR22}$ mutant neurons and a *tau* transgene in *shot*$^3$ mutant neurons. In both cases, Syt staining revealed a rescue of the synapse reduction phenotype (*Figure 1—figure supplement 2*). Taken together, our results indicate that Shot and Tau functionally overlap, rather than act hierarchically in the same pathway.

Next, we investigated synaptic phenotypes in vivo. Since *shot*$^3$ and *tau*$^{MR22}$ animals are late embryonic lethal, we analysed them at late embryonic stage 16, when Syt is already confined to nascent synaptic terminals, as can be reliably imaged at neuromuscular junctions (NMJs; *Figure 2* and *Figure 2—figure supplement 1* for a schematic drawing of the embryonic NMJ) (*Littleton et al., 1993*). In *shot-tau* mutant embryos, Syt levels at NMJs were reduced to 48%, whereas *shot* mutant embryos showed a milder reduction to 71%, and *tau* mutant embryos no detectable effect (*Figure 2*). Taken together, our data suggest that Tau is required for the formation of synapses in culture and in vivo and that Tau and Shot functionally overlap in this context.

## Synapse maintenance in the ageing brain requires Tau and Shot

Tau and Shot remain highly expressed in mature neurons (see later in Figure 6), and we tested whether they are required also for synapse maintenance. For this, we used the GAL4-UAS system to co-express previously used and validated *UAS*-RNAi constructs for both genes in the same neurons (*Bolkan and Kretzschmar, 2014*; *Subramanian et al., 2003*). This strategy takes out Tau and Shot functions with some delay, due to the late onset of GAL4 expression and the persistence of Tau and Shot proteins (*Figure 3—figure supplement 1*).

We first used this approach in cultured primary neurons, where combined knock-down of *tau* and *shot* caused no reduction in the number of Syt-labelled presynaptic sites at 3 and 18 DIV as compared to wildtype controls (*Figure 3A–B*), indicating normal synapse development. However, at 26 DIV, Syt puncta in knock-down neurons were reduced to 41% (*Figure 3A–B*), which was comparable to the *shot-tau* double-mutant phenotype at 2 DIV (*Figure 1*). At all time points (i.e. 3, 18 and 26 DIV), there were no measurable changes in axonal length nor in branch number when compared to control neurons (*Figure 3—figure supplement 4*) clearly indicating that the strong reduction in Syt positive synapses in 26 DIV knock-down neurons was not a secondary effect of morphological changes such as in number of branches and axonal length (*Figure 3A and B*).

To assess roles in synapse maintenance also in vivo in the ageing brain, we used *atonal-Gal4 (ato-Gal4)* to drive gene expression in dorsal cluster (DC) neurons of the adult brain (*Zschätzsch et al., 2014*) (illustrated in *Figure 3—figure supplement 2*). In these experiments, we expressed GFP-tagged Synaptotagmin (Syt-GFP) to label synapses, either alone or together with *shot*$^{RNAi}$ and/or *tau*$^{RNAi}$. We compared young flies at 2–5 days after eclosion with old flies at 24–29 days at 29oC. We found that the number of Syt-GFP labelled synapses in DC neurons decreased to 34% in aged specimens expressing both *shot*$^{RNAi}$ and *tau*$^{RNAi}$ when normalised to young flies of the same genotype (*Figure 3C–D*). This age-dependent decrease in synapse numbers did not occur in control flies (*Figure 3C–D*), and single knock-down of either *shot* or *tau* only showed a non-significant tendency to lose synapses over time (*Figure 3D*). Notably, aged double knock-down DC neurons had no reduction in the number of axonal branches (as assessed with the myr-tdTomato membrane marker; *Figure 3E–F*), indicating that also in vivo precocious synapse decay was not due to axonal loss.

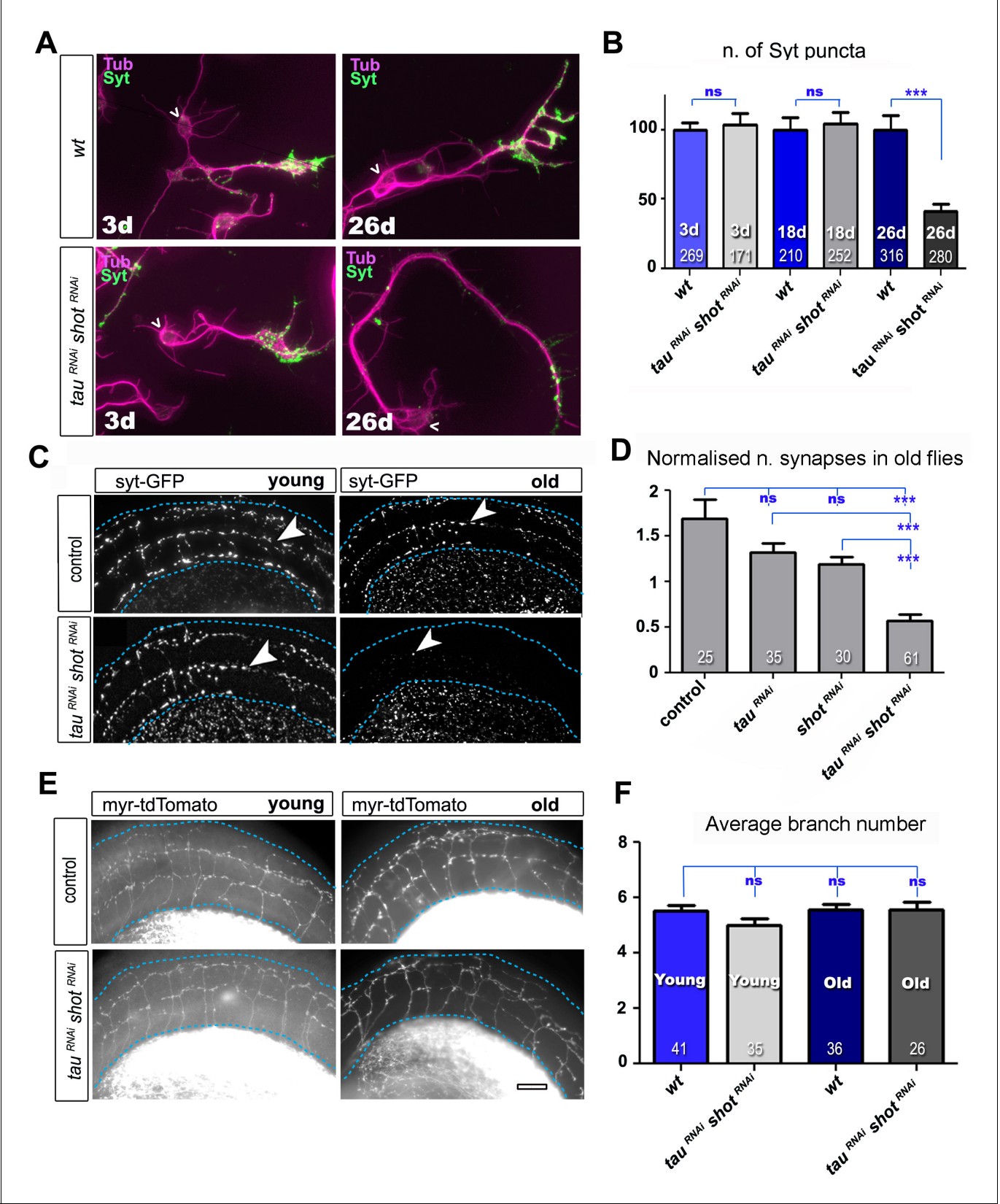

**Figure 3.** Tau and Shot are required for the maintenance of synaptic markers in cultured neurons and the ageing adult fly brain. (**A**) Primary neurons at 3 DIV and 26 DIV cultured from embryos that were wildtype or jointly expressing *UAS-tau*[RNAi] and *UAS-shot*[RNAi] in all neurons driven by the pan-
*Figure 3 continued on next page*

*Figure 3 continued*

neuronal driver *elav-Gal4 (tau^RNAi shot^RNAi)*. Neurons are stained with anti-tubulin and anti-Syt; at 26 DIV, *tau^RNAi shot^RNAi* neurons display a reduction in the number of Syt puncta when compared to wildtype. (B) Quantification of the experiments in A, shown as the number of Syt puncta per neuron at 3 DIV, 18 DIV and 26 DIV, normalised to wildtype controls (the number of assessed neurons is indicated in each bar; ***$P_{MW}$<0.001; ns, not significant $P_{MW}$>0.05). (C) A region of *Drosophila* adult brains including the medulla (delimited by dashed lines) where Syt-GFP is expressed in dorsal cluster neurons using *atonal*-Gal4, in the absence (control) or together with *tau^RNAi* and *shot^RNAi (tau^RNAi shot^RNAi)*. Brains are stained with anti-GFP at 2–5 days (young) and 24–29 days (old) after eclosion. Note that GFP-labelled synapses (arrowheads) are decreased in old brains upon *shot* and *tau* knock-down. (D) Quantification of the experiments in C, showing the normalised number of Syt-GFP-labelled puncta in old specimen per mean number of puncta in young specimens for the following phenotypes: *ato-Gal4 UAS-syt-GFP* alone (*control*), co-expressing *UAS-tauRNAi (tau^RNAi)*, *UAS-shot^RNAi(shot^RNAi)*, or both knock-down constructs (*tau^RNAi shot^RNAi*; the number of analysed brains is indicated in each bar, ***$P_{MW}$<0.001; ns, not significant $P_{MW}$>0.05). (E) Brain regions as in C, of animals expressing the membrane marker myr-tdTomato driven by *ato-Gal4* revealing the morphology of the projections of dorsal cluster neurons within the medulla ; brains were from adults at 2–5 days (young) and 24–29 days (old) after eclosure, expressing myr-tdTomato either alone (control) or together with *tau^RNAi* and *shot^RNAi(tau^RNAi shot^RNAi)*. (F) Quantification of the experiments in E, displayed as number of branches per axon projecting into the medulla (the number of axons analysed is indicated in each bar; ns, not significant $P_{MW}$>0.05). Scale bar: 10 μm in A and 40 μm in C and E. A statistics summary of the data shown here is available in *Figure 3—source data 1*.

The following source data and figure supplements are available for figure 3:

**Source data 1.** Summary of the statistics from *Figure 3B,D,F*.

**Figure supplement 1.** Delayed effect of RNAi mediated knock-down of Shot and Tau.

**Figure supplement 1—source data 1.** Summary of the statistics from *Figure 3—figure supplement 1B*.

**Figure supplement 2.** Schematic drawings of brain areas analysed in this study.

**Figure supplement 3.** Loss of function mutations in *shot* and *tau* induce morphological changes.

**Figure supplement 4.** RNAi-mediated knock-down of Shot and Tau has no effect on axonal length and branch number.

**Figure supplement 4—source data 1.** Summary of the statistics from *Figure 1—figure supplement 3C and D*.

From our studies in culture, in embryos and in the adult brain, we conclude that Tau and Shot are required for synapse development during early stages, and for synapse maintenance in ageing neurons, where their combined deficiency causes precocious synapse loss.

## Tau and Shot control intracellular trafficking of synaptic proteins

Synapse formation and maintenance require that synaptic proteins synthesised in the soma are actively transported through the axon towards the distant presynaptic sites. In *Drosophila* primary neurons, transport of endogenous synaptic proteins already starts at 8 hr in vitro (HIV) when synaptic proteins appear as dotted patterns along axons and in growth cones (*Sánchez-Soriano et al., 2010*) (*Figure 4A–B*). This is similar in rat hippocampal neurons (*Bonanomi et al., 2005*). Already at this early stage, *shot-tau* double mutant neurons display a strong decrease in synaptic proteins in growth cones and axons (*Figure 4A–B*), indicating potential intracellular transport defects.

To study intracellular transport, we analysed the dynamics of Syt-GFP using live imaging of neurons at 8 HIV. In *shot-tau* mutant neurons, the percentages of anterograde and retrograde displacements and retrograde velocities of Syt-GFP containing vesicles were not affected and anterograde velocities were only slightly increased. In contrast, the number of Syt vesicles within the axon showed a sharp decrease to ~40% in *shot-tau* mutant neurons when compared to controls (*Figure 4C*). Notably, this decrease in axonal vesicles is accompanied by an increase in the number of somatic Syt-GFP puncta to ~159% (*Figure 4C*). Similarly, endogenous Syt was increased in somata of *shot-tau* mutant neurons, both in culture and in in vivo (*Figure 4D–E*).

These phenotypes in *shot-tau* mutant neurons suggested aberrant intracellular trafficking of Syt-containing vesicles, potentially due to a road block in the soma.

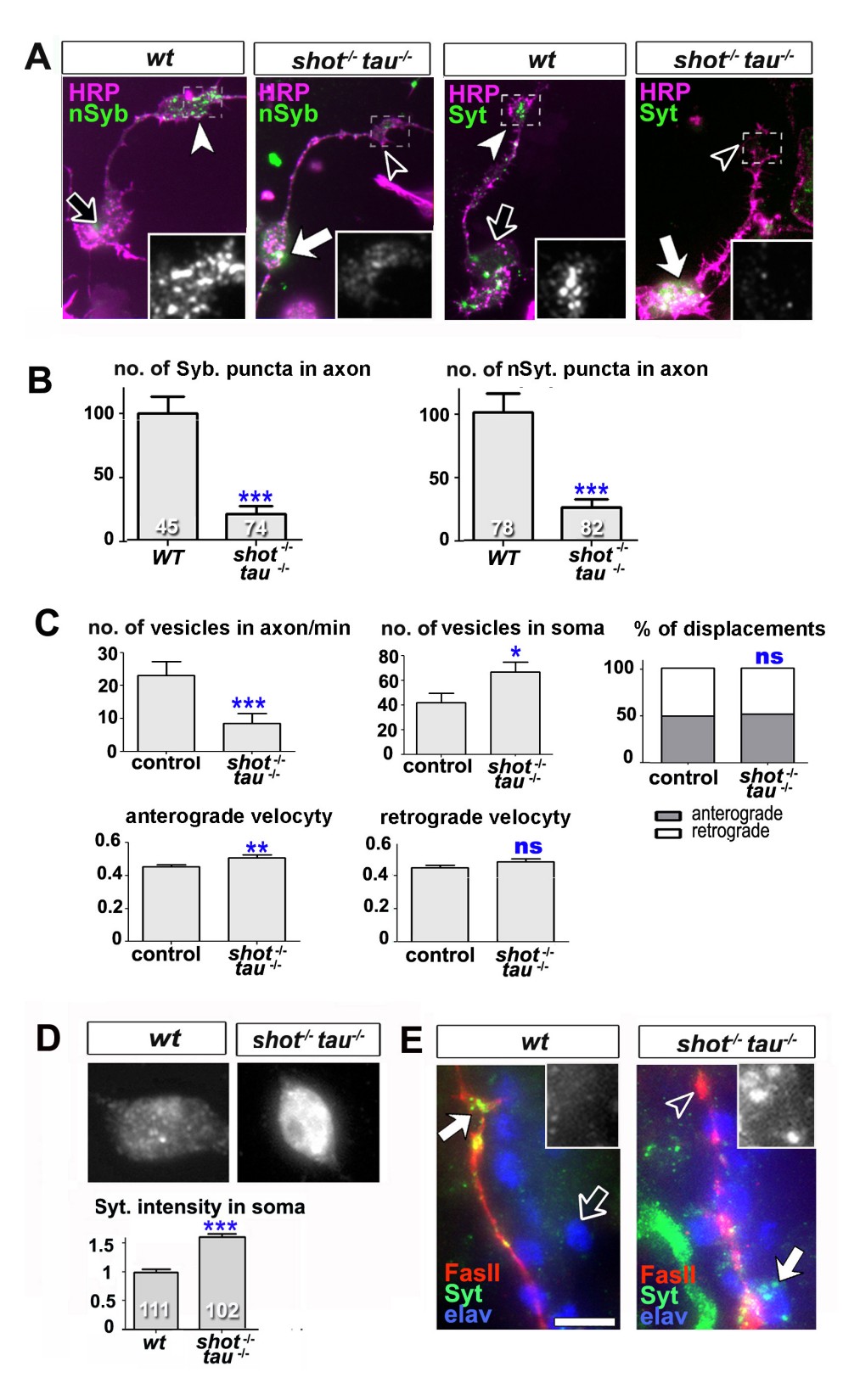

**Figure 4.** Intracellular transport of synaptic proteins is defective in *shot-tau* mutant neurons. (**A**) Primary *Drosophila* neurons at 8HIV, obtained from embryos that were wildtype (wt) and *shot-tau* (*shot⁻ᐟ⁻ tau⁻ᐟ⁻*) stained with antibodies against pan-neuronal HRP (magenta), Syt (green) or nSyb (green); *Figure 4 continued on next page*

*Figure 4 continued*

nSyb and Syt are reduced in the growth cones (open versus white arrow heads) but enriched in cell bodies (open versus white arrows) of *shot-tau* mutant neurons. (**B**) Quantification of the experiments from A, given as the number of nSyb or Syt puncta in axons and growth cones; the number of analysed neurons is given in the bars (***$P_{MW}$<0.001). (**C**) Quantification of various transport parameters generated from live movies of axons of wildtype or *shot-tau* mutant neurons (*shot^{-/-} tau^{-/-}*) at 8 HIV with *elav-Gal4* driven expression of *UAS-Syt-GFP*. Axonal anterograde and retrograde velocities show only subtle or no alteration in the axons of *shot-tau* neurons. On the contrary, the numbers of vesicles in axons of *shot-tau* neurons are sharply decreased and increased in the somata (**$P_{MW}$<0.01; *$P_{MW}$<0.05; ns, not significant $P_{MW}$>0.05). (**D**) Magnified views of the somata from primary *Drosophila* neurons at 2 DIV, obtained from wildtype (wt) and *shot-tau* mutant embryos (*shot^{-/-} tau^{-/-}*), co-stained with antibodies against Syt. To document the protein content within cell bodies, several z stacks per neuron were obtained and fused as maximal projection; the cell bodies show higher levels of Syt in *shot-tau* mutant neurons as compared to wildtype (number of assessed cells is indicated in the bars, average staining intensity normalised to wildtype; ***$P_{MW}$<0.001). (**E**) The dorsal peripheral nervous system (PNS) of wildtype and *shot-tau* embryos at late stage 16 (stages according to) (*Campos-Ortega and Hartenstein, 1997*) stained for Syt (green), FasII (red) and the pan-neuronal nuclear marker Elav (blue). The nascent NMJ at the tip of the inter-segmental motornerve (red) in wildtype contains high levels of Syt (white arrow) whereas the somata of sensory neurons (blue and grey in insets) contain low levels (open arrow); in *shot-tau* homozygous embryos the somata of sensory neurons have high levels of Syt (arrow and inset), whereas there is only little staining at the nerve tip (open arrowhead). Scale bars: 10 µm in A, 5 µm in D and 5 µm in E. A statistics summary of the data shown here is available in *Figure 4—source data 1*.

The following source data is available for figure 4:

**Source data 1.** Summary of the statistics from *Figure 4B–D*.

## Tau and Shot regulate the activity of kinesin-3

Type 3 kinesins are the predominant motors driving axonal transport of synaptic proteins (*Hirokawa et al., 2010*). This is also the case for the *Drosophila* homologue Unc-104 (also called Imac) (*Pack-Chung et al., 2007*). We found that *unc-104^{170}* null mutant primary neurons at 2 DIV have a vast reduction of Syt-stained synapses (*Figure 5A*). This phenotype is strikingly similar to the one observed in *shot-tau* mutant neurons, and suggested that Shot-Tau might regulate Unc-104 function.

To test this hypothesis, we performed genetic interaction studies. We found that primary neurons stained for Syt at 2 DIV and heterozygous for all of the three genes (*shot^{-/+} unc-104^{-/+} tau^{-/+}*) displayed significant reduction in the number of Syt-stained synapses when compared to heterozygous condition of the *unc-104* or *shot-tau* mutant alleles alone (*Figure 5A*). Also triple-heterozygous mutant embryos at late stage 16 displayed reduced Syt staining at neuromuscular terminals, but increased staining in the cell bodies of CNS and sensory neurons (*Figure 5C*; see *Figure 2—figure supplement 1* for a schematic drawing of the embryonic NMJ and CNS). Therefore, *unc-104^{170}* mutant, *shot-tau* mutant, and triple-heterozygous mutant neurons all show similar phenotypes, both in culture and in vivo, suggesting a functional link between these three proteins.

Type 3 kinesins are anterograde motor proteins that move towards axon tips in mouse neurons (*Niwa et al., 2013*), and we also find *Drosophila* Unc-104 to be distally enriched in the axons of primary neurons at 2 DIV and in embryonic motorneurons *in vivo* (*Figure 5D–E*). In mouse, this distal localisation was shown to be suppressed when blocking kinesin-3 mobility (*Niwa et al., 2013*). Also in *shot-tau* mutant neurons in culture and in vivo, Unc-104 localisation in distal axons is reduced whereas levels in the somata are increased (*Figure 5D–E*), suggesting that insufficient amounts of Unc-104 move from the somata into axons.

To test whether diminished Unc-104 levels in axons are the cause for the synaptic defects in *shot-tau* mutant conditions, we over-expressed Unc-104, which fully restored synapse numbers in *shot-tau* mutant neurons at 2 DIV (*Figures 5B versus 1D*). Notably, Unc-104 over-expression in *shot-tau* mutant neurons achieved this rescue of synapses in neurons with significantly less axonal branches (*Figure 3—figure supplement 3F*), clearly demonstrating that both features are regulated independently of each other.

We next examined whether Unc-104 plays comparable roles also during synapse maintenance in the ageing brain. We used the *ato-Gal4* driver (*Zschätzsch et al., 2014*) and co-expressed Unc-104 together with *shot^{RNAi}* and *tau^{RNAi}*. To label synapses we expressed the presynaptic marker neuronal Synaptobrevin-GFP (nSyb-GFP, due to technical reasons the use of nSyb-GFP was more convenient than Syt-GFP). Consistent with our previous findings with Syt-GFP (*Figure 3C–D*), also nSyb-GFP

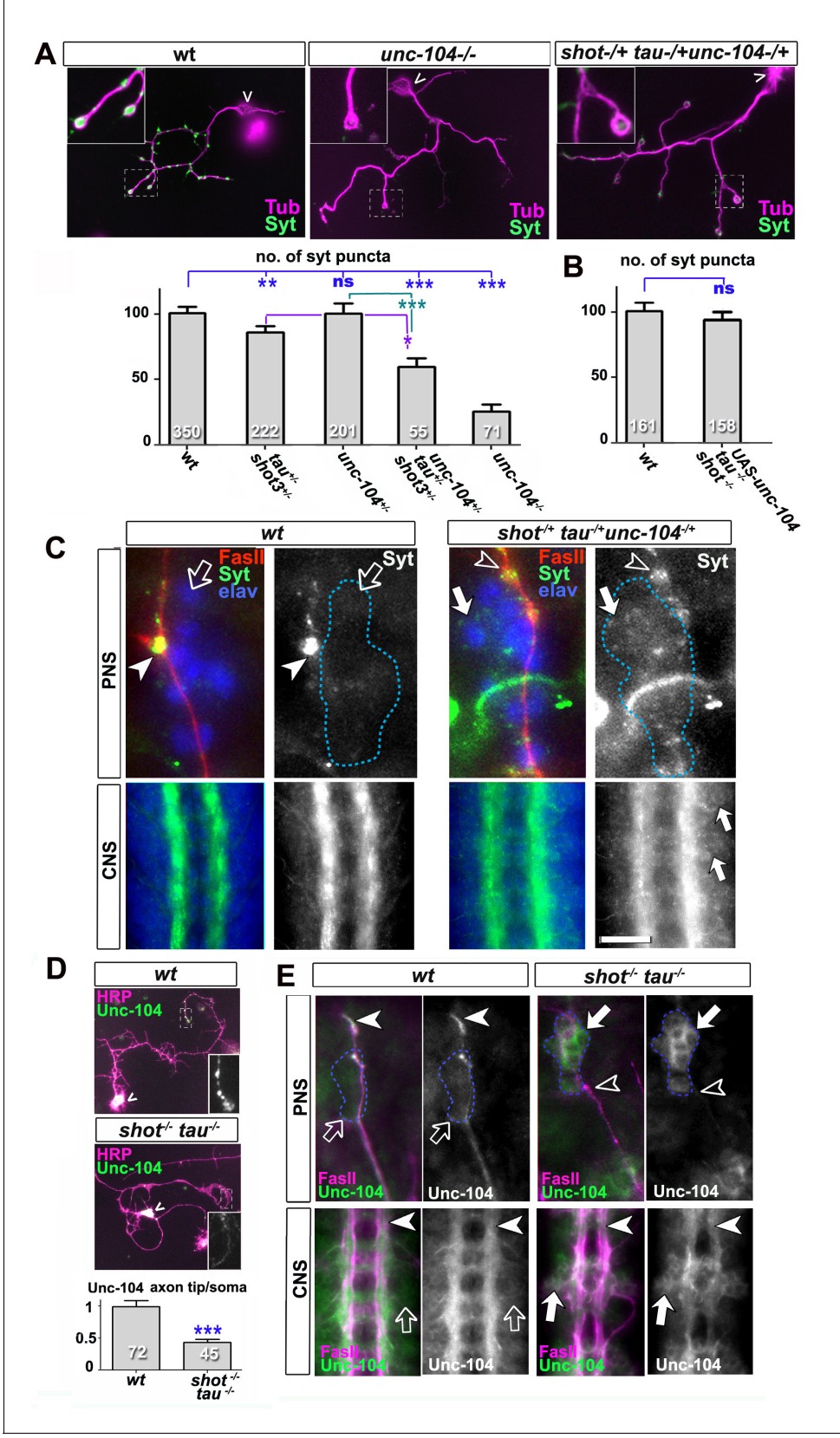

**Figure 5.** Defects in kinesin-3 function mediate synaptic deficits in *shot-tau* mutant neurons. Shot and Tau interact with Unc-104 and regulate its subcellular distribution. (**A**) Primary *Drosophila* neurons at 2 DIV, obtained from embryos which were wildtype, homozygous for *unc104imac170* (*unc104-/-*), *Figure 5 continued on next page*

*Figure 5 continued*

or triple-heterozygous for *shot³ tau^{MR22} unc104^{imac170}* mutations (shot^{+/-} tau^{+/-}unc104^{+/-}), co-stained with antibodies against tubulin (Tub, magenta) and Syt (green). The graph shows the quantification of the data including also *unc104^{-/+}* and *shot^{-/-} tau^{-/-}* controls. (B) Quantification of Syt puncta in two day old neurons, obtained from embryos that were wildtype or *shot^{-/-} tau^{-/-}* with *elav-Gal4* driven expression of *UAS-unc-104* (compare **Figure 1D**). (C) The dorsal peripheral nervous system (PNS) and the central nervous system (CNS) of wildtype and *shot³ tau^{MR22} unc104^{imac170}* triple heterozygous embryos at late stage 16 (stages according to **Campos-Ortega and Hartenstein, 1997**) stained for Syt (green), FasII (red in upper panel) and the pan-neuronal nuclear marker Elav (blue); for illustration of the imaged tissue see **Figure 2-figure supplement 1**. The nascent NMJ at the tip of the inter-segmental motornerve (red in upper panels) in wildtype contains high levels of Syt (arrowheads) whereas the somata of sensory neurons (blue; demarcated by dashed lines) contain low levels (open arrows); in *shot³ tau^{MR22} unc104^{imac170}* triple heterozygous embryos the somata of sensory neurons have high levels of Syt (arrows), whereas there is only little staining at the nerve tip (open arrowhead). In the ventral nerve cord of wildtype (lower panels), Syt is confined to the neuropile (synapse containing CNS compartment; arrowheads) and excluded from the cortex (compartment with the cell bodies of inter- and motorneurons); in the ventral nerve cord of *shot³ tau^{MR22} unc104^{imac170}* triple heterozygous embryos, there are segmental groups of cell bodies displaying higher Syt levels (arrows). (D) Primary *Drosophila* neurons at 2 DIV, obtained from wildtype (wt) and *tau-shot* mutant embryos, stained with antibodies against pan-neuronal HRP (magenta) and Unc-104 (green); Unc-104 in distal axon segments (emboxed and magnified in insets) is enriched in wildtype but much weaker in *shot-tau* mutant neurons (chevrons indicate neuronal somata). Data were quantified as average intensity of Unc-104 at the distal end of the axon divided by the average intensity at the soma. (E) Upper and lower panels show the same locations of late stage 16 embryos as shown in C, but taken from wildtype and *shot-tau* mutant embryos, stained for FasII (magenta) and Unc-104 (green). Note the stark decrease of Unc-104 at the end of motor nerves (open versus white arrow heads) and the unusual accumulations of Unc-104 in the cell bodies of sensory neurons as well as in the CNS cortex in *shot-tau* embryos (open versus white arrows). In all graphs, the number of assessed neurons is indicated in each bar; ***$P_{MW}$<0.001; *$P_{MW}$<0.05; ns, not significant $P_{MW}$>0.05; scale bars: 18 μm in A, 5 μm in C/PNS, 35 μm in C/CNS, 15 μm in D and E/PNS, 35 μm in E/CNS. A statistics summary of the data shown here is available in **Figure 5—source data 1**.

The following source data and figure supplements are available for figure 5:

**Source data 1.** Summary of the statistics from **Figure 5**.

**Figure supplement 1.** Expression of Unc-104 rescues synaptic defects in aged adult brains.

**Figure supplement 1—source data 1.** Summary of the statistics from **Figure 5—figure supplement 1B**.

revealed age-dependent synapse reduction upon *shot^{RNAi}* and *tau^{RNAi}* expression, clearly confirming our previous data (**Figure 5—figure supplement 1** and **8E–F**). When Unc-104 was co-expressed, synapse reduction was clearly rescued (Figure 5—figure supplement 1).

Taken together, our data are consistent with a model where Shot-Tau loss generates a road block which inhibits Unc-104 translocation from the soma into axons, causing synaptic defects at developmental stages and in ageing neurons.

## Loss of Shot-Tau induces microtubule destabilisation accompanied by changes in JNK activation

To address the mechanistic links from loss of Shot-Tau to aberrant transport and synaptic defects, we focussed on microtubules. Shot localises along microtubules, and *shot* mutant neurons treated with the microtubule-destabilising drug nocodazole display unusual gaps in their axonal microtubule bundles (**Figure 6B–C**) (**Alves-Silva et al., 2012**; **Sanchez-Soriano et al., 2009**). Tau also localises along microtubules (**Figure 6A**, **Video 1**), and *tau^{MR22}* mutant neurons likewise displayed axonal microtubule gaps upon nocodazole treatment which could be rescued with targeted expression of Tau (**Figure 6B–C**). Both, *shot* and *tau* mutant neurons treated with nocodazole displayed on average one gap per axon. This number is significantly increased to ~3 gaps in *shot-tau* mutant neurons (**Figure 6B–C**), demonstrating that Tau and Shot share a common function in microtubule stabilisation.

To test whether their roles in microtubule stabilisation and synapse regulation are linked, we treated *shot-tau* mutant embryos at early stage 16 for 3 hr with the microtubule-stabilising drug epothilone B (**Goodin et al., 2004**). We found a significant rescue of Syt levels at motoraxonal endings, which was not observed in vehicle-treated controls (**Figure 6D–E**). Therefore, a decrease in microtubule stability is a likely cause for defective transport of synaptic proteins in *shot-tau* mutant neurons. It could be argued that Shot-Tau dependent microtubule stabilisation directly regulates processive advance of kinesins in axons (see Discussion), yet the rather normal transport dynamics

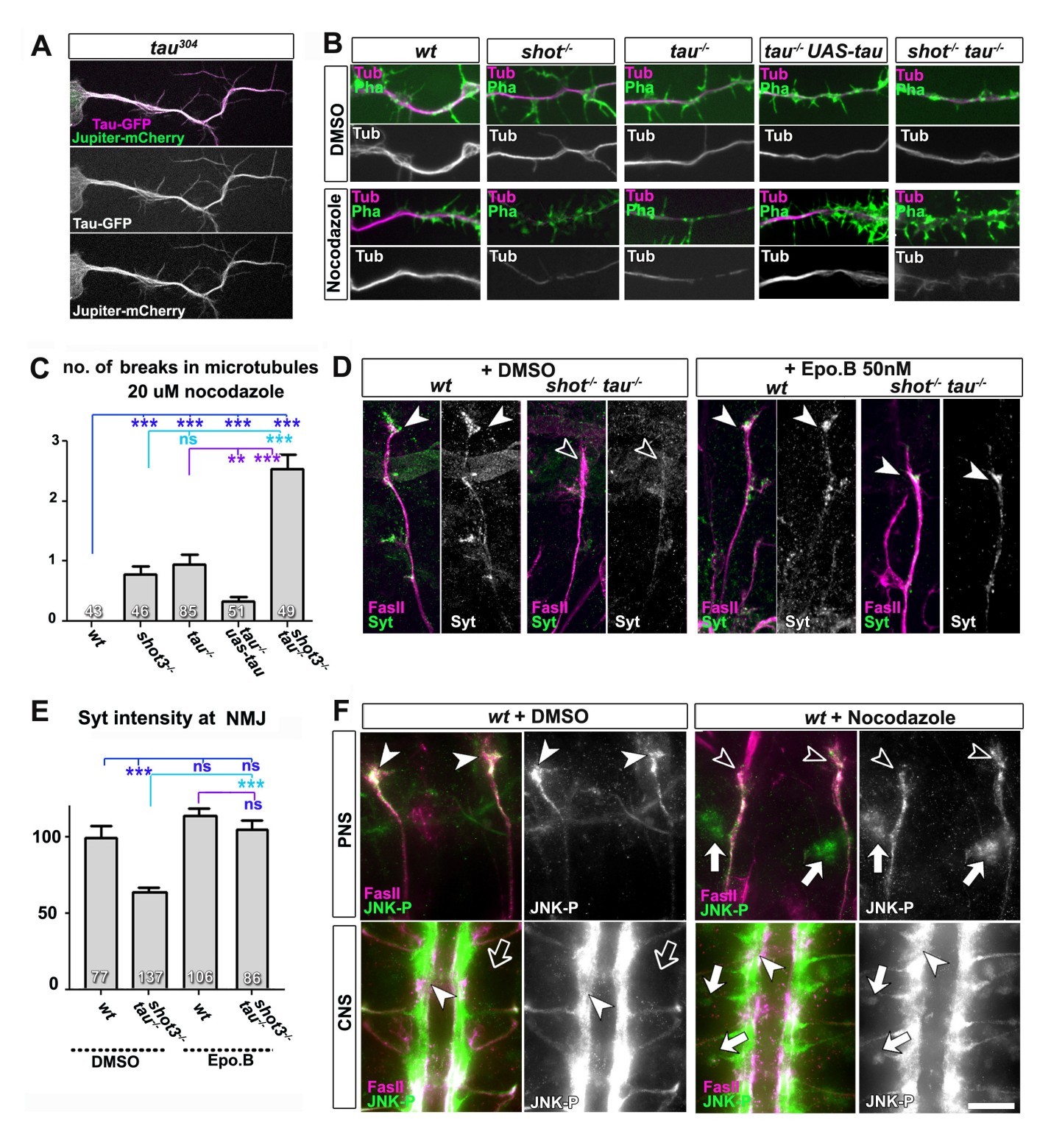

**Figure 6.** Microtubule instability mediates aberrant JNK signalling and synaptic defects. (**A**) Live imaging of *Drosophila* neurons at 2 DIV, obtained from embryos carrying *tau304* (a protein trap line where the endogenous tau gene is genomically tagged with GFP) and the microtubule binding protein Jupiter-Cherry. Endogenous Tau (in magenta) is observed in a pattern reminiscent of microtubules, and colocalises with Jupiter (shown in green). (**B**) Axons of *Drosophila* neurons at 6 HIV with the following genotypes: wildtype (*wt*), *shot3* (*shot-/-*), *tauMR22* (*tau-/-*), tau rescue (*tau-/- UAS-tau*) and *shot-tau* (*shot-/- tau-/-*). Neurons were treated for 2.5 hrs with vehicle (DMSO) or 20 µM nocodazole, fixed and stained with anti-Tubulin (Tub, magenta and white) *Figure 6 continued on next page*

*Figure 6 continued*

and phalloidin (Pha, green). Only *shot³*, *tau^MR22*, and *shot-tau double mutant* displayed gaps in their axonal microtubule bundles upon nocodazole treatment, but not wildtype and *tau* mutant embryos with Tau rescue. (C) Quantification of the experiments in B, indicated as the number of breaks in the microtubule staining per axon (number of analysed neurons is indicated in bars; ***$P_{MW}<0.001$; **$P_{MW}<0.01$; ns, not significant $P_{MW}>0.05$). (D) Embryonic motoraxons of wildtype and *shot-tau* embryos at late stage 16 treated with vehicle (DMSO) or 50 nM of the microtubule stabilising drug epothilone B for 3 hr and stained with FasII (magenta) and Syt (green); in wildtype, the nascent NMJ at the nerve tip contains high levels of Syt (arrowheads); in *shot-tau* embryos there is only little Syt staining at the nerve tip (open arrowhead). Treatment of *shot-tau* embryos with 50nM epothilone B increases the levels of Syt at the tip of motornerves (arrowheads). (E) Quantification of the experiments shown in D, measured as the average intensity of Syt at nascent NMJs and normalised to wildtype (number or assessed NMJ is indicated in bars; ***$P_{MW}<0.001$; ns, not significant $P_{MW}>0.05$). (F) Upper (PNS) and lower (CNS) panels show the same locations of late stage 16 wildtype embryos as shown in *Figure 5C*, stained for FasII (magenta) and activated phospho-JNK (JNK-P); treatment with 100 μm nocodazole for 2 hrs induced a relocation of JNK-P from nascent NMJs (open versus white arrow heads) to cell bodies of sensory neurons and in the CNS cortex (white versus open arrows). Scale bar: 5 μm in A, 4 μm in B, 10 μm in E, 15 μm in D/PNS and 35 μm in D/CNS. A statistics summary of the data shown here is available in *Figure 6—source data 1*.

The following source data and figure supplements are available for figure 6:

**Source data 1.** Summary of the statistics from *Figure 6C and E*.

**Figure supplement 1.** Treatment of *shot-tau* mutant neurons with epothilone B, increases the localisation of JNK-P at axonal tips.

**Figure supplement 1—source data 1.** Summary of the statistics from *Figure 6—figure supplement 1B*.

we observed upon live imaging in *shot-tau* mutant neurons clearly excluded this possibility (*Figure 4C*).

Instead, we hypothesised that microtubule aberration indirectly promotes a transport roadblock in somata. As a potential mediator, we suspected the JNK signalling pathway which is known to respond to a number of cellular stresses (see Discussion). To test our hypothesis, we investigated the pattern of JNK activity, using an antibody against phosphorylated JNK (JNK-P) (*Langen et al., 2013*). In wild type embryos at stage 16, we found high accumulations of JNK-P at motoraxon tips and low levels in the somata of CNS and sensory neurons (*Figures 6F* and *7A*), i.e. a localisation pattern similar to that of synaptic proteins and Unc-104 (*Figure 5C–E*). This distribution was altered in single *tau^MR22* or *shot³* mutant embryos, showing higher levels of JNK-P in neuronal somata and lower levels at the tips of motoraxons (*Figure 7A*). This altered pattern was intensified in *shot-tau* double mutant neurons (*Figure 7A*) and clearly reminiscent of the redistribution patterns observed with synaptic proteins and Unc-104 in these neurons (*Figures 4E* and *5E*). Notably, these changes in the pattern of JNK activation were reproduced when inducing microtubule stress by applying nocodazole to early stage 16 wildtype embryos (*Figure 6F*). Complementary to this finding, treatment of *shot-tau* mutant neurons with the microtubule stabilising drug epothilone B, increased the localisation of JNK-P at axonal tips and reduced the aberrant localisation in somata (*Figure 6—figure supplement 1A,B*).

These data suggested a cascade of events where *shot-tau* mediated microtubule destabilisation or stress triggers abnormal JNK activation in somata which, in turn, causes a somatic block of Unc-104 mediated synaptic transport. In strong support of this hypothesis, the three key players of this cascade, JNK-P, Unc-104 and synaptic proteins, show a striking correlation by concentrating unanimously at axon tips in wildtype, but in somata in *shot-tau* mutant neurons (*Figures 4E*, *5E* and *7A*).

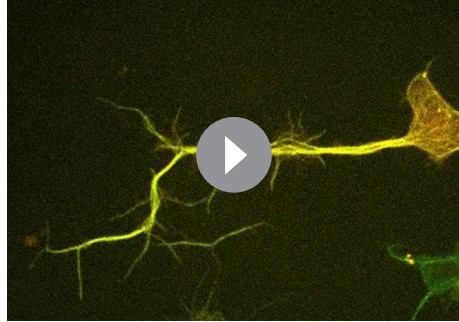

**Video 1.** Live imaging of *Drosophila* neurons at 2 DIV, obtained from embryos carrying *tau^304* (a protein trap line where the endogenous tau gene is genomically tagged with GFP) and the microtubule binding protein Jupiter-Cherry. Endogenous Tau (in green) is observed in a pattern reminiscent of microtubules, and colocalises with Jupiter (shown in red). The time laps were obtained every 15 s with a 3i Marianas Spinning Disk Microscope.

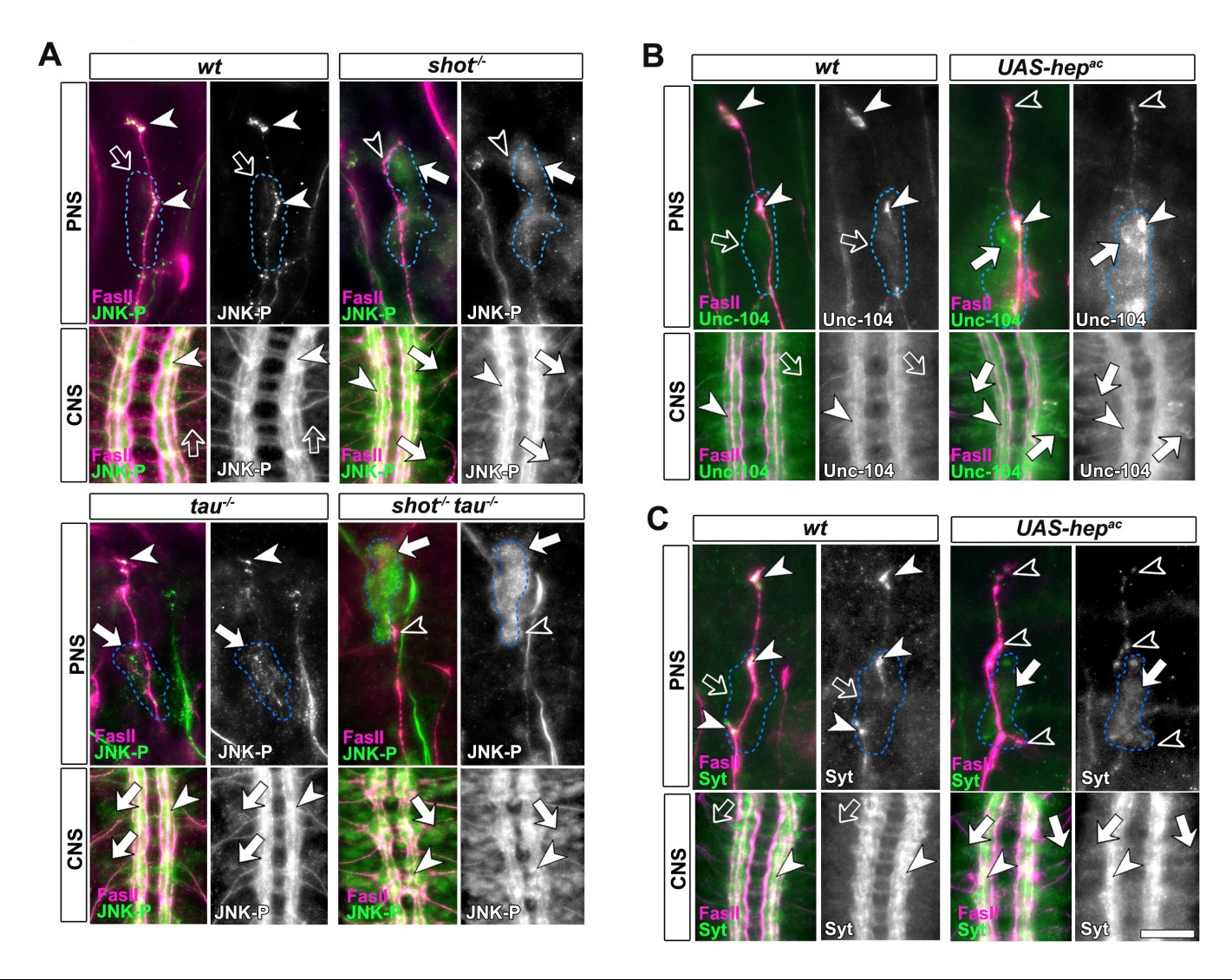

**Figure 7.** Activated JNK correlates with the subcellular localisation of Unc-104 and Syt. Upper (PNS) and lower (CNS) panels in A-C show the same locations of late stage 16 embryos as shown in *Figure 5E*, but embryos are of different genotypes and stained with different antibodies, as indicated; genotypes: wildtype (*wt*), *shot³* (*shot⁻/⁻*), *tau^{MR22}* (*tau⁻/⁻*), *shot-tau* (*shot⁻/⁻ tau⁻/⁻*), *elav-Gal4* driven expression of *UAS-hep^{ac}* (*UAS-hep^{ac}*); used antibodies detect FasII (magenta), Syt (green), Unc-104 (green), activated phospho-JNK (JNK-P). (**A**) In wildtype, JNK-P is high at nerve endings (white arrow heads) and below detection levels in cell bodies of sensory neurons and in the CNS cortex (open arrows); this pattern is inverted in *tau^{MR22}* and *shot³* mutant embryos and even stronger in *shot-tau* embryos, i.e. Syt is reduced at nerve endings (open arrowheads) and upregulated in cell bodies (white arrows). (**B, C**) Artificial activation of JNK with neuronal expression of Hep^{ac} suppresses high levels of Unc-104 and Syt at nascent NMJs (open versus white arrow heads) and increases their levels in cell bodies (white versus open arrows). Scale bars: 15 μm in PNS panels and 35 μm in CNS panels.

## Aberrant JNK signalling upon Shot-Tau loss causes the somatic road-block of Kinesin-3 transport and synaptic defects

To prove that JNK acts downstream of *shot-tau* to regulate Unc-104, we first expressed a constitutively active variant of the MAPKK Hemipterous (Hep^{AC}), a known activator of the JNK pathway (*Glise et al., 1995*). In late stage 16 embryos, indiscriminate JNK activation through Hep^{AC} triggered an accumulation of Unc-104 and Syt in somata and a decrease of both proteins at axon tips (*Figure 7B–C*). Also in primary mature neurons at 2 DIV, Hep^{AC} caused a reduction in the number of synapses to 43% (*Figure 8A–B*). Therefore, Hep^{AC} expression mimicked the defects observed in

*shot-tau* mutant neurons, consistent with a model where aberrant JNK pathway activation upon Shot-Tau loss causes the somatic block of Unc-104-dependent synaptic transport.

If our model is correct, attenuation of the JNK pathway should rescue the synaptic defects in *shot-tau* mutant neurons. To downregulate the JNK pathway, we used loss-of-function of the JNK activating kinase Wallenda/DLK (*wnd²*) (*Valakh et al., 2013*) and overexpression of the JNK inhibiting phosphatase Puckered (Puc) (*Martin-Blanco et al., 1998*). When combined with the *tau^{MR22}* mutation, both genetic tools for JNK downregulation fully rescued the synaptic defects in primary neurons at 2 DIV (*Figure 8A–B*). Even more, *wnd²* fully rescued synapse reduction in *shot-tau* double mutant neurons at 2 DIV (*Figure 8A–B*), and Syt levels at NMJs of *shot-tau* mutant embryos in vivo (*Figure 8C–D versus Figure 2*).

So far, our data suggest that JNK acts downstream of *shot-tau* to regulate Unc-104. In this case, attenuation of the JNK pathway should also rescue aberrant Unc-104 localisation in *shot-tau* mutant neurons. Accordingly, *wnd²* restored correct localisation of Unc-104 in *shot-tau* double mutant neurons at 2 DIV (*Figure 8—figure supplement 1*), and in *shot-tau* mutant embryos *in vivo* (*Figure 8—figure supplement 2B versus Figure 5D,E*).

Having confirmed JNK as the essential mediator of *shot-tau* synaptic defects, we tested whether it acts through the canonical pathway involving the AP1 transcription factor (*Ciapponi and Bohmann, 2002*), or by phosphorylating other targets in the cytoplasm. For this, we used a well established LOF mutant allele of the *kayak/c-fos* gene (*kay²*) which removes one constituent of the AP1 heterodimer and mimics various known JNK mutant phenotypes (*Ciapponi and Bohmann, 2002*). Unlike *wnd²* or Puc overexpression, the *kay²* mutation failed to rescue the synaptic phenotypes of *tau^{MR22}* in primary neurons (*Figure 8A–B*). This strongly suggests that the JNK pathway inhibits synaptic transport by acting independently of AP1 dependent transcription.

In conclusion, the JNK pathway is both required and sufficient to mediate between Shot-Tau loss and their downstream synaptic phenotypes in developing neurons by causing a transport roadblock, and this likely occurs through phosphorylating cytoplasmic targets in the soma.

## JNK mediates ageing related synaptic decay caused by Shot-Tau loss

To test whether JNK plays comparable roles also during synapse maintenance in the ageing brain, we used the *ato-Gal4* driver (*Zschätzsch et al., 2014*) and co-expressed a dominant negative variant of the *Drosophila* JNK homolog Basket (*bsk^{DN}*) (*Adachi-Yamada et al., 1999*) together with *shot^{R-NAi}*, *tau^{RNAi}* and nSyb-GFP. We found that co-expression of *bsk^{DN}* was able to rescue the synapse reduction phenotype (*Figure 8E,F*), thus confirming JNK as a mediator between the effects of *shot-tau* and precocious synapse decay also in ageing neurons (summarised in *Figure 9*).

## Discussion

### A new mechanism of synaptic pathology caused by loss of Tau and Shot

The aim of our studies was to understand the role of endogenous Tau in neurons with particular attention to synapses. This effort was essentially aided by our finding that Tau and Shot are functionally redundant, and the subsequent incorporation of Shot into our studies. The robust phenotypes of *shot-tau* double-mutant neurons enabled us to demonstrate roles of Shot-Tau during the formation and maintenance of pre-synaptic sites in axons, and unravel the underlying mechanistic cascade which involves three major steps. Firstly, the absence of Shot-Tau causes microtubule destabilisation. Secondly, this cytoskeletal stress causes aberrant JNK activity patterns with upregulation in somata and downregulation at axon tips. Thirdly, aberrant JNK activation leads to a somatic roadblock for kinesin-3 mediated transport, thus inhibiting the delivery of synaptic proteins and eventually causing synapse loss. Depending on whether the functions of Tau and/or Shot are removed during development or ageing, either the formation or the maintenance of synapses are affected, respectively (*Figure 9*).

Our model explaining the function of Tau and Shot in synapse establishment and maintenance by regulating intracellular transport, is supported by loss- and gain-of-function experiments, genetic interactions and cross-rescue experiments. The initial finding that *shot-tau* mutant neurons had reduced branch numbers, could have suggested that defects on synapse numbers is indirect. However, experiments with double knock-down in culture and in the adult brain clearly showed strong

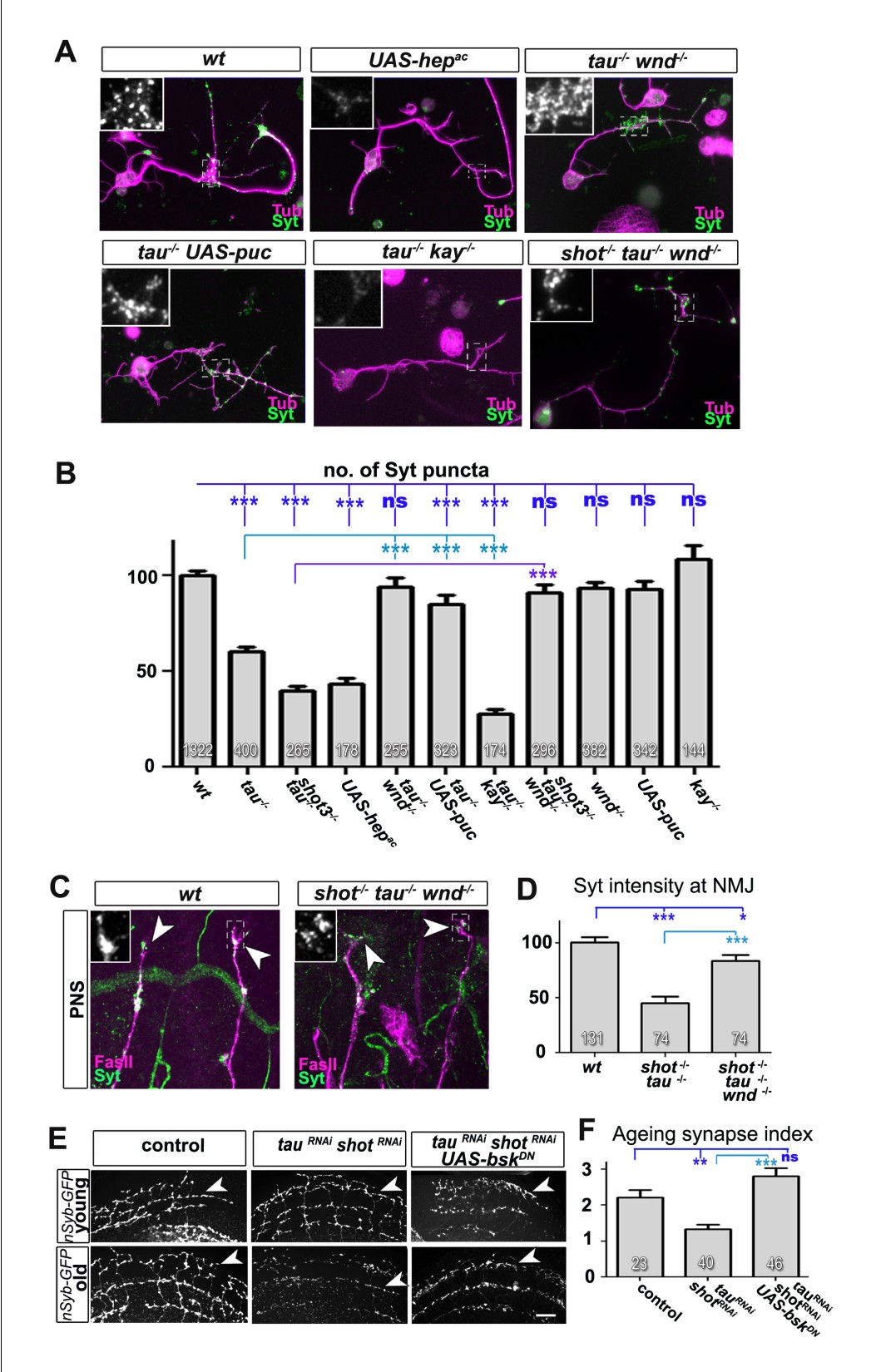

**Figure 8.** Inhibition of the JNK pathway rescues synaptic defects in *shot-tau* mutant neurons. (**A**) Primary *Drosophila* neurons at 2 DIV, obtained from embryos of the following genotypes: wildtype (wt), *elav-Gal4* driven expression of *UAS-hep^ac* (UAS-hep^ac), *tau^MR22* (tau^-/-), *wnd^2* (wnd^-/-), *tau^-/-* with *elav-*
*Figure 8 continued on next page*

Figure 8 continued

Gal4 driven expression of *UAS*-puc (tau^-/- UAS-puc), *tau^MR22 kay^2* (tau^-/- kay^-/-) and *shot^3 tau^MR22 wnd^2* (shot^-/- tau^-/- wnd^-/-), all stained with antibodies against Tubulin (tub, magenta) and Syt (green). Insets correspond to emboxed areas and show a magnified view of the Syt staining. (B) Quantification of experiments in A, shown as the number of Syt puncta normalised to wildtype (number of assessed neurons is shown in the bars, ***$P_{MW}$<0.001; **$P_{MW}$<0.01; *$P_{MW}$<0.05; ns, not significant $P_{MW}$>0.05). (C) Inter-segmental motornerves in the dorsal area of wildtype and *shot^3* mutant embryos at late stage 16, stained against FasII (magenta) and Syt (green); insets correspond to emboxed areas and show a magnified view of the most dorsal nascent NMJs stained for Syt; note the rescue of Syt localisation if Wnd is absent in *tau-shot* mutant background. (D) Quantification of the experiments in C, measured as the average intensity of Syt normalised to wt (number of assessed NMJs is shown in the bars; ***$P_{MW}$<0.001; *$P_{MW}$<0.01). (E) A region of *Drosophila* adult brains including the medulla; UAS-nSyb-GFP is expressed in dorsal cluster neurons using *atonal*-Gal4, either alone (control), together with tau^RNAi and shot^RNAi (*tau^RNAi shot^RNAi*) or together with tau^RNAi, shot^RNAi and UAS-bsk^DN. Brains are shown at 2–6 days (young) and 26–30 days at 29°C after eclosion (old); GFP-labelled synapses are decreased in old brains with *shot-tau* knock-down when compared to controls, and this effect is rescued by the expression of Bsk^DN. (F) Quantification of experiments in E, shown as number of GFP-labelled synapses in old specimen per mean number of GFP-labelled synapses in young specimens of the respective genotype (number of analysed brains is indicated in the bars; ***$P_{MW}$<0.001; **$P_{MW}$<0.01). Scale bars: 5 µm in A, 10 µm in C and 40 µm in E. A statistics summary of the data shown here is available in *Figure 8—source data 1*.

The following source data and figure supplements are available for figure 8:

**Source data 1.** Summary of the statistics from *Figure 8B,D and F*.

**Figure supplement 1.** Attenuation of the JNK pathway rescues aberrant Unc-104 localisation in *shot-tau* mutant neurons in culture.

**Figure supplement 1—source data 1.** Summary statistics from *Figure 8—figure supplement 1B*.

**Figure supplement 2.** Attenuation of the JNK pathway rescue aberrant unc-104 localisation in *shot-tau* mutant embryos.

synapse reduction whilst maintaining normal branch patterns, and Unc-104 rescued synapse reduction in *shot-tau* mutant neurons without major increases of the branch pattern in these neurons. These results clearly demonstrate that changes in neuronal morphology are not the cause of changes in synapse number.

Notably, the synaptic function of Tau described here for *Drosophila* might be conserved in higher animals or humans, since also aged Tau knock-out mice develop a reduction of synaptic proteins from the hippocampus (*Ma et al., 2014*).

## Implications of our findings for Tau-related pathologies

Our findings provide potential new mechanistic explanations for various tau related brain disorders. For example, microdeletions in the region of MAPT (the human tau gene) cause intellectual disability (*Sapir et al., 2012*), and Tau's synapse-promoting roles m ay well contribute to this pathology. Furthermore, various tauopathies are characterised by precocious pathological loss of synapses. Our data suggest that loss of tau could lead to defective synapse maintenance and eventually synapse loss. For example, a prominent group of dementias which lacks distinctive histopathology (DLDH) are characterised by the loss of Tau (*Zhukareva et al., 2001*). Further tauopathies including Alzheimer disease, typically involve hyper-phosphorylation and aggregate formation of Tau (*Hernández and Avila, 2007*; *Williams, 2006*). In this scenario, there are two parallel, non-exclusive modalities through which Tau can cause pathology. Firstly, detached hyper-phosphorylated tau attains gain-of-function roles in the cytoplasm damaging neurons through a number of mechanisms (*Morris et al., 2013*). Secondly, hyper-phosphorylation of tau causes a loss-of-function condition by depleting Tau from microtubules. However, since Tau knock-out mouse models mostly failed to show significant phenotypes and the neuronal functions of endogenous tau remain little understood, the pathological importance of Tau loss from microtubules has been marginalised (*Morris et al., 2013*). Our results now re-emphasise the notion that loss of Tau from microtubules could contribute to neurodegenerative pathology and deliver mechanistic explanations.

To unravel pathomechanisms caused by the loss of Tau, we mostly used combined depletion of Shot and Tau, which gave us strong phenotypes, ideal for short-term experimental approaches. However, we found similar, yet milder phenotypes if only Tau was depleted, suggesting that the mechanisms described here could well contribute to slow disease progression in tauopathies. Our discovery that spectraplakins are MAPs which functionally overlap with Tau, opens up new

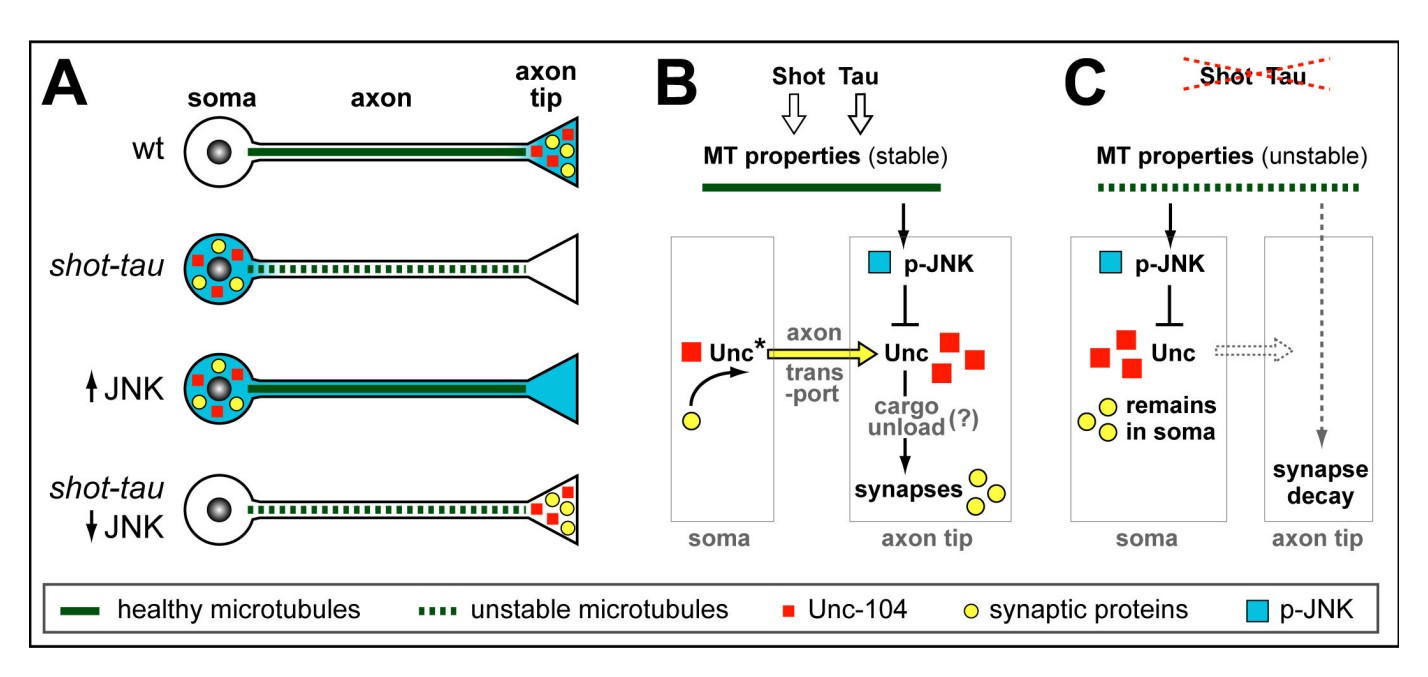

**Figure 9.** Schematic model of proposed function for Tau and Shot. (**A**) Neurons illustrating different phenotypes: in wildtype neurons (wt), microtubules are stable (green line) and levels of Unc-104 (red square), synaptic markers (yellow dots) and p-JNK (turquoise background) are high in axon tips; in shot-tau mutant neurons, microtubules are unstable (stippled green lines), and the above listed proteins accumulate in cell bodies (soma); ubiquitous activation of JNK (↑JNK) causes similar somatic accumulation of Unc-104 and synaptic markers; down-regulation of JNK (↓JNK) rescues the shot-tau mutant phenotypes. (**B,C**) Schematic representation of the underlying mechanisms: In wildtype neurons (**B**), Unc-104 is activated (Unc*) and mediates axonal transport of synaptic proteins (yellow arrow) to the axon tip, where we propose (?) that active JNK inhibits Unc-104, thus releasing its cargo for synaptic incorporation. In shot-tau mutant neurons (**C**), unstable microtubules cause upregulation of JNK in the soma, thus inhibiting Unc-104 and trapping it as well as its cargo proteins in the soma.

experimental avenues for Tau studies. So far, spectraplakins have been linked to the degeneration of sensory and autonomous neurons (*Edvardson et al., 2012*; *Ferrier et al., 2013*), and it remains to be elucidated whether they may have similar roles also in the brain. Our results clearly hint at this possibility.

## A novel mechanism for Tau-dependent regulation of neuronal transport

The loss of Tau and/or Shot inhibits kinesin-3 mediated transport leading to accumulation of synaptic proteins in the soma of neurons. We propose a road-block mechanism suppressing the initiation of axonal transport in somata of Shot-Tau depleted neurons, which is caused indirectly through microtubule stress and mediated by JNK (*Figure 9*).

The involvement of microtubules in causing a transport block is supported by our experiments using microtubule stabilising and de-stabilising drugs which rescued or mimicked the shot-tau mutant phenotypes, respectively. Similarly, axonal transport defects and cognitive deficits of *PS19Tg mice* (expressing the P301S mutant form of human *tau)* and various other mouse and fly tauopathy models were shown to be rescued by microtubule-stabilising drugs (*Gozes, 2011*; *Quraishe et al., 2013*; *Shemesh and Spira, 2011*; *Zhang et al., 2012*), suggesting that the mechanisms we described may be conserved and relevant to disease.

The somatic road-block is a novel mechanism through which the loss of Tau can interfere with the transport of synaptic proteins and provides potential explanations also for somatic accumulations of postsynaptic proteins such as PSD-95, AMPA and NMDA receptors observed in mouse tauopathy models (*Hoover et al., 2010*; *Shao et al., 2011*). A likely mechanism causing a roadblock in intracellular transport could be the direct inactivation of Unc-104 or its associated adaptor proteins, for example through JNK or other kinases within its pathway. This mode of regulation has a clear

precedent in kinesin-1 and its adaptor Jip which are directly phosphorylated by JNK leading to transport inhibition (*Stagi et al., 2006*). Unfortunately, our extensive attempts to co-immunoprecipitate JNK and Kinesin-3 were unsuccessful (data not shown), leaving open for now the exact molecular mechanism.

## JNK is an important mediator between *shot-tau* induced microtubule stress and synapse loss

We propose that aberrant JNK activation downstream of microtubule destabilisation or stress is the ultimate cause for the defective delivery of synaptic proteins in Tau and/or Shot loss of function. Also in mouse, microtubule stress leads to somatic activation of the JNK pathway, suggesting this mechanism is likely to be conserved with vertebrates (*Valakh et al., 2015*).

The JNK pathway is emerging as a central player in neurodegenerative diseases. Its activation is prompted by various neurodegeneration risk factors including oxidative stress, inflammation, and ageing (*Lotharius et al., 2005*; *Valakh et al., 2015*). Furthermore, JNK is activated in AD patients (*Coffey, 2014*) and in several AD models where it triggers progression of the pathology (*Sclip et al., 2014*). The new link between Tau/spectraplakins, JNK and synapses we propose here, is therefore likely to provide mechanistic explanations for synaptic pathology observed in AD and other tauopathies.

## Conclusions

We have delivered an important conceptual advance by revealing a new mechanistic cascade which can explain synaptic decay as the consequence of Tau loss from microtubules. Furthermore, we identified a previously unknown functional redundancy with spectraplakins as a promising new avenue for research on Tau. Our findings emphasise that Tau detachment from microtubules can be an important aspect contributing to the pathology of tauopathies in parallel to roles of hyper-phosphorylated Tau in the cytoplasm. Synaptic decay, axonal transport and alterations in the JNK pathway are emerging as central players in a wider range of adult-onset neurodegenerative diseases, and here we have aligned these factors into a concrete mechanistic cascade.

# Materials and methods

## Fly stocks

The following fly stocks were used: the Gal4 driver lines *sca-Gal4* (*Sánchez-Soriano et al., 2010*), *elav-Gal4* (3$^{rd}$ chromosome) (*Luo et al., 1994*) and *ato-Gal4* (*Zschätzsch et al., 2014*); the mutant alleles *Df(3R)tauMR22* (*Bolkan and Kretzschmar, 2014*; *Doerflinger et al., 2003*), *shot$^3$* (*Kolodziej et al., 1995*), *unc-104$^{imac170}$* (courtesy of Dr. T. Schwarz) (*Pack-Chung et al., 2007*), *wnd$^2$* (*Collins et al., 2006*) and *kay$^2$* (*Ciapponi and Bohmann, 2002*) (the latter two courtesy of S. Sweeney); the UAS lines UAS-*tau-GFP* (*Doerflinger et al., 2003*), UAS-*shot-GFP* (*Alves-Silva et al., 2012*; *Sanchez-Soriano et al., 2009*). *tau$^{GD25023}$* (UAS-tau$^{RNAi}$; Vienna *Drosophila* RNAi Center, Austria) (*Bolkan and Kretzschmar, 2014*), UAS-shot$^{RNAi}$ (*Subramanian et al., 2003*), UAS-syt-*GFP* (3$^{rd}$ and 2$^{nd}$ chromosome, Bloomington Stock Center), UAS-*nSyb-GFP* (Bloomington Stock Center), UAS-*tdTomato* (*Zschätzsch et al., 2014*), tau$^{304}$ (Bloomington Stock Center), Jupiter-Cherry (Bloomington Stock Center), UAS-*Hep-ac* (*Glise et al., 1995*), UAS-*bsk$^{DN}$* (*Adachi-Yamada et al., 1999*) and UAS-*puc* (*Martin-Blanco et al., 1998*) (the latter five fly stocks courtesy of B. Hassan). Lethal fly stocks were kept over balancers carrying *twist-Gal4* and *UAS-GFP* constructs (*Halfon et al., 2002*), and combinations of mutant alleles and transgenic constructs were generated using conventional genetic crosses (*Prokop, 2013*).

## Cell culture

The generation of primary neuronal cell cultures was described previously (*Prokop et al., 2012*; *Sánchez-Soriano et al., 2010*). In brief, to generate *Drosophila* primary cultures, neurons were extracted from stage 11 embryos (*Campos-Ortega and Hartenstein, 1997*). Whole embryos were treated for 1 min with bleach to remove the chorion, sterilized for ~30 s in 70% ethanol, washed in sterile Schneider's/FCS, and eventually homogenized with micro-pestles in 1.5 ml tubes containing about 21 embryos per 100 µl dispersion medium. This was followed by 4–5 min incubation in dispersion

buffer containing collagenase and dispase at 37°C, followed by a wash in sterile Schneider's/FCS and eventually resuspension in the final volume of Schneider's medium. Cells were plated onto coverslips coated with 0.5 mg/ml Concanavalin A (Sigma) and kept as hanging drop cultures in air-tight special culture chambers (*Küppers-Munther et al., 2004*) usually for 8 hr, 2–3, 18 or 26 days at 26°C. Dilutions of the MT destabilising drug nocodazole (20 µM; Sigma) in Schneider's medium were prepared from stock solutions in DMSO. For controls, equivalent concentrations of DMSO were diluted in Schneider's medium.

## Drug treatment of *Drosophila* embryos

Stage 16 embryos were dissected flat in D ulbecco's Phosphate Buffered Saline (*Budnik et al., 2006*) and cultured for several hours in Schneider's medium with or without drugs. Dilutions of the microtubule destabilising drug nocodazole (20 µM; Sigma) and the microtubule stabilizer epothilone B (50 nM; Sigma) in Schneider's medium were prepared from stock solutions in DMSO. For controls, equivalent concentrations of DMSO were diluted in Schneider's medium.

## Immunohistochemistry

Primary fly neurons were fixed in 4% paraformaldehyde (PFA) in 0.1 M phosphate buffer (PB; pH 7–7.2) for 30 min at room temperature (RT). Stage 16 embryos were dissected flat in Dulbecco's Phosphate Buffered Saline (*Budnik et al., 2006*) and fixed with 4% PFA for 30 min. Adult fly brains were dissected in Dulbecco's Phosphate Buffered Saline and fixed with 4% PFA for 15 min. Antibody staining and washes were performed with Phosphate Buffered Saline supplemented with 0.3% Triton X-100. Staining reagents: anti-Tubulin (clone DM1A, mouse, Sigma; alternatively, clone YL1/2, rat, Millipore Bioscience Research Reagents); anti-FasII (clone ID4, mouse, DSHB, RRID: AB_532376); anti-GFP (goat, Abcam RRID: AB_305643); Cy3/FITC-conjugated anti-HRP (goat, Jackson ImmunoResearch); anti-Syn (SYNORF1 3C11, mouse, DSHB, RRID:AB_528479); anti-Brp (DSHB, RRID:AB_2314867); anti-Syt (rabbit, was a gift from Dr. S. Sweeney); anti-nSyb and anti-Unc104 (both rabbit, were a gift of Dr. T. Schwarz); anti-Elav (rat, DSHB, RRID:AB_528218); anti-pJNK (rabbit, pTPpY, Promega, RRID:AB_430864), anti-CD2 (mouse, AbD Serotec, RRID:AB_566608), anti-dTau (Nick Lowe), anti-Shot (Talila Volk) FITC-, Cy3- or Cy5-conjugated secondary antibodies (donkey, purified, Jackson ImmunoResearch). Specimens were embedded in Vectashield (VectorLabs).

## Microscopy and data analysis

Standard documentation was performed with AxioCam monochrome digital cameras (Carl Zeiss Ltd.) mounted on BX50WI or BX51 Olympus compound fluorescent microscopes. Z-stacks of embryonic CNSs were taken with a Leica DM6000 B microscope and extracted with Leica MM AF Premier software. Z-stacks of adult fly brains were taken with a Leica DM6000 B microscope or with a 3i Marianas Spinning Disk Confocal Microscope. Using custom software written in Python and NumPy, fly brain images taken with a Leica DM6000 B microscope were individually band-pass filtered (A trous wavelet [1][2], linear 3x3 filter, keeping scales 1–4) to remove stationary background.

To quantify the number of synaptic densities in mature neurons in culture and the number of vesicles containing synaptic proteins in 8h neurons in culture, we used ImageJ (RRID:SCR_003070). In detail, we used thresholding to select synaptic densities from axons of single isolated cells, followed by particle analysis. For all experiments done in parallel, identical thresholds were used. For the quantification of synapses in mature neurons in culture, we selected polarised neurons with a clear distinguishable axon, the same neurons were used to study axon length and number of branches.

To quantify synaptic proteins or Unc-104 in the soma of neurons, we manually selected the area of the somata using the tubulin or HRP channel and measured the signal intensity derived from the Syt or Unc-104 channel. To measure the levels of Unc-104 at the tip of axons, we selected an area of the same size at the most distal part of axons and measured the signal intensity derived from the Unc-104 channel. To quantify synaptic proteins at the tip of embryonic motorneurons in vivo. we manually selected the area occupied by the growth cones using the FasII staining and measured the signal intensity derived from the Syt channel; the background intensity was subtracted. Images used for these measurements did not contain saturated levels. Also, to measure the number of synaptic densities in DC neurons in the medulla of the adult brain, we used thresholding to select synaptic

densities followed by particle analysis. The number of branches in the medulla per DC neuron was quantified manually. To quantify MT stability upon nocodazole treatment, we counted the number of breaks in the microtubule bundle per axon.

Time lapse imaging of cultured primary neurons (in Schneider's/FCS) was performed on a Delta Vision RT (Applied Precision) restoration microscope using a 100x/1.3 Ph3 Uplan Fl objective and the Sedat filter set (Chroma 89000). The images were collected using a Coolsnap HQ (Photometrics) camera. The temperature control was set to 26°C. For time lapse recording, images were taken every 2 s for 2 min. To generate transport measurements, vesicles containing fluorescently tagged Syt were tracked manually using the manual tracking plugin for ImageJ.

## Statistics, replicates and sample-size

All data are shown as mean with SEM. Statistical analyses were performed in GraphPad Prism using Mann-Whitney Rank Sum Tests (indicated as $P_{MW}$) or $Chi^2$ ($P_{Chi}$), with 95% confidence intervals. The exact p-values and sample size are indicated in the figure legends.

For each primary neuronal cell culture experiment (technical replication), approximately 30 to 40 embryos were used. Neurons obtained from those embryos were divided and cultured in 3 to 4 independent chambers (biological replication). The sample size provided corresponds to the number or neurons studied. Most experiments were performed at least 2 times (2 technical repeats) meaning a minimum of 60 embryos were used, and a minimum of 6 independent culture chambers were studied. These experiments are shown in *Figure 1B* (*tau*$^{-/-}$, 4 technical replications, 11 biological replications), *Figure 1D* (Syt, *shot*$^{-/-}$, 3 technical replications, 9 biological replications; *shot*$^{-/-}$ *tau*$^{-/-}$, 3 technical replications, 9 biological replications), *Figure 1D* (Brp *shot*$^{-/-}$, 2 technical replications, 8 biological replications; *tau*$^{-/-}$ 2 technical replications, 6 biological replications), *Figure 3B* (all genotypes at each time point 2 technical replications, 6–8 biological replications), *Figure 4D* (all genotypes 2 technical replications, 6 biological replications), *Figure 5B* (2 technical replications, 6 biological replications), *Figure 8B* (*tau*$^{-/-}$ *Uas-Puc*, 3 technical replications, 8 biological repeats; *tau*$^{-/-}$ *wnd*$^{-/-}$, 2 technical replications, 6 biological repeats; *shot*$^{-/-}$ *tau*$^{-/-}$ *wnd*$^{-/-}$, 2 technical replications, 6 biological replications). Other experiments were as follows: *UAS-tau* rescue experiment of synaptic defects (*Figure 1B*) and microtubule stability defects (*Figure 6C*) in *tau*$^{-/-}$ were performed with 1 technical replication which included at least 30 embryos distributed in 4 and 2 independent chambers or biological replications, respectively. However, for these particular experiments we used a co-culture technique in which *tau*$^{-/-}$ control neurons were cultured alongside with *tau*$^{-/-}$ UAS-Tau neurons and therefore are subject to the same environmental variations. For *shot-tau* mutant neurons stained with Brp (*Figure 1D*) we used 1 technical replication which included at least 30 embryos and 3 independent biological replications. For *tau*$^{-/-}$ *kay*$^{-/-}$ and *kay*$^{-/-}$ (*Figure 8B*), we used 1 technical replication which included at least 30 embryos and 5 independent biological replications. For the measurement of Syt and nSyb synaptic puncta in 8 HIV neurons (*Figure 4B*), we performed 1 technical replication for each synaptic protein, which included at least 30 embryos each and 2 independent biological replications. For the measurement of Unc-104 in 2 day neurons (*Figure 5D*) we performed 1 technical replication which included at least 30 embryos and 45 neurons were measured. To account for variations in the immunohistochemistry procedure, we calculated the ratio between distal axon and soma.

For the quantification of axonal transport (*Figure 4C*), we performed 2 technical replications which included at least 30 embryos each, 2 independent biological replications from which 10–14 neurons were analysed. For measurements of Syt in the nascent embryonic NMJ (*Figure 2B* and *8C*) we used at least 15 embryos per genotype (biological replication) and performed at least 2 technical replications. Both controls and mutant embryos were dissected and stained in the same chamber and therefore subjected to equal conditions (*Figure 2B* *shot*$^{-/-}$ 3 technical replications, 30 biological replications, *tau*$^{-/-}$ 2 technical replications, 15 biological replications, *shot*$^{-/-}$ *tau*$^{-/-}$ 3 technical replications, 17 biological replications; *Figure 8C* *shot*$^{-/-}$ *tau*$^{-/-}$ *wnd*$^{-/-}$ 2 technical replications, and 15 biological replications). For treatment of embryos with epothilone B (*Figure 6E*) we used at least 12 embryos per genotype and performed 2 technical replications. Both controls and *shot-tau* mutant embryos were present in the same treatment chambers and therefore subjected to equal conditions.

For the study of synaptic phenotypes in adult brains (*Figure 3D*), we performed at least 3 technical replications and used a minimum of 30 brains in total per genotype. For the quantification of axonal branches in adult brains we performed at least 2 technical replications and we used a minimum

of 11 brains. For rescues of synaptic phenotypes in adult brains with UAS-*bsk*<sup>DN</sup> (*Figure 8F*) we performed 3 technical replications and used a minimum of 40 brains in total.

## Acknowledgements

This work was made possible through funding by the BBSRC (BB/M007456/1) to NSS, by the BBSRC (BB/I002448/1) and Wellcome Trust ISSF (105610/Z/14/Z) to AP, and the German Science Foundation (DFG; VO 2071/1-1) to AV. The Bioimaging Facility microscopes used in this study were purchased with grants from the BBSRC, The Wellcome Trust and the University of Manchester Strategic Fund, and the Manchester Fly Facility where flies were kept and genetic crosses performed has been supported by funds from The University of Manchester and the Wellcome Trust (087742/Z/08/Z). We thank Meredith Lees and Egor Zindy for experimental help, Bassem Hassan, Roland Brandt and Nigel Hooper for helpful comments on the manuscript, Thomas Schwarz, Sean Sweeney, Bassem Hassan, Nick Lowe and Talila Volk for kindly providing reagents. Stocks obtained from the Bloomington *Drosophila* Stock Center (NIH P40OD018537) were used in this study.

## Additional information

### Funding

| Funder | Grant reference number | Author |
| --- | --- | --- |
| Biotechnology and Biological Sciences Research Council | BB/M007456/1 | Natalia Sanchez-Soriano |
| Biotechnology and Biological Sciences Research Council | BB/I002448/1 | Andreas Prokop |
| Wellcome Trust | 105610/Z/14/Z | Andreas Prokop |
| Deutsche Forschungsge-meinschaft | 2071/1-1 | Andre Voelzmann |

The funders had no role in study design, data collection and interpretation, or the decision to submit the work for publication.

### Author contributions

AV, Acquisition of data, Analysis and interpretation of data, Drafting or revising the article; PO-R, YQ, MC-M, MdC-E, Acquisition of data, Analysis and interpretation of data; AP, Conception and design, Drafting or revising the article; NS-S, Conception and design, Acquisition of data, Analysis and interpretation of data, Drafting or revising the article, Contributed unpublished essential data or reagents

### Author ORCIDs

Andre Voelzmann, http://orcid.org/0000-0002-7682-5637
Pilar Okenve-Ramos, http://orcid.org/0000-0002-7513-6557
Yue Qu, http://orcid.org/0000-0002-2593-3654
Andreas Prokop, http://orcid.org/0000-0001-8482-3298
Natalia Sanchez-Soriano, http://orcid.org/0000-0002-6667-2817

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
