## [Decision Letter]

Thank you for submitting your article "Tau and spectraplakins promote synapse formation and maintenance through JNK and neuronal trafficking" for consideration by *eLife*. Your article has been reviewed by two peer reviewers, and the evaluation has been overseen by a Reviewing Editor and K VijayRaghavan as the Senior Editor. The following individuals involved in review of your submission have agreed to reveal their identity: Sean Sweeney (Reviewer #2).

The reviewers have discussed the reviews with one another and the Reviewing Editor has drafted this decision to help you prepare a revised submission.

Summary:

Voelzmann et al. provide a compelling set of data that gives a deep insight into tau function with regard to axon traffic with development and age, by pinpointing a redundancy with short-stop/spectraplakin, and a pathway with kinesin-3/immaculate-connections. They use an elegant series of experiments involving primary culture, in vivo experiments, genetics, pharmacology, ageing and live imaging to demonstrate this. As the discussion states, this data has profound implications for our understanding of tau function in neurodevelopment and neurodegeneration. Yet, the authors try to cover a large area for which they need to support numerous potential causalities.

While the overall idea is strong, there are numerous open ends and insufficiently supported statements in this manuscript to support the cascade-like chain of causalities presented here.

Essential revisions:

The abstract and introduction might be toned down a little and move away from the idea of 'homeostasis'. This term has become all things to all folks, and 'synaptic maintenance' or similar phrasing could be used. The authors should take time within the results to explain the experimental system – i.e. the tau mutants are lethal, and the primary culture allows examination of neurons beyond this lethal stage. For Figure 1, we would suggest including, or referring to the later figure where double heterozygotes of shot/tau give significant data – this more directly alludes to the pathway, rather than two loss-of-function mutants making a defective phenotype worse (But see below on this). They also refer often to 'synapses' when a more precise description of (something like) 'materially deficient' synapses would be useful. In general it will be useful to make the statements precise and clear. The diagrammatic explanations of the experimental system are beautifully clear, as are the images presented.

The strongest experiments in the paper are the rescue of tau, shot mutants through unc104 overexpression and through JNK attenuation. However, in these cases, only selective read-outs are used (for example, we do not learn whether JNK attenuation affects unc104, as predicted by the model). Sometimes read-outs focus on syt punctae, sometimes brp, n-syb and different ages in culture or in vivo. Different read-outs in different experiments make comparisons more difficult. A principal concern arising from the presentation of selective read-outs is the specificity of the phenotypes. Defective syt puncta (the most commonly used read-out in the paper) can arise directly and indirectly from numerous causes. Just because they become more when there is more kinesin does not mean that the function of tau and shot is truly rescued. I am wondering what these neurons look like, what other markers look like and what aspects of neuronal function and maintenance are truly 'rescued' with unc104 overexpression compared to the microtubule-stabilizing drug compared to JNK signaling attenuation. Finally, a key issue is the lack of simple genetic rescue experiments. The authors only provide one tau rescue experiments at the beginning of the paper. The shot mutant is never rescued by a shot transgene.

In sum, while we find the paper compelling and interesting substantial revisions are required and should be done within the two month period required by *eLife*.

Required experiments:

1) Rescue of shot, rescue attempt of tau with shot and rescue attempt of shot with tau. Only these experiment reveal functional overlap. We do not think the rescue of the double mutant with either tau or shot is as meaningful, because the double mutant clearly gets worse than any single and any single rescue in that background would make it better, without revealing functional overlap of the two proteins.

2) Morphology of the neurons throughout, in mutants and rescues, to show to what extent syt puncta counts are a consequence of more severe neuronal defects.

3) Further regarding the specificity issue: Please see the reference in the review below to how the original unc104 paper made its point on specificity (mitochondria, HRP, fasII, mCD8, – subset of these). The key is whether this is a defect specific to synaptic proteins as proposed or a general transport or even health problem of these neurons.

4) The authors should use UNC-104 as a readout in the rescues and throughout for mutants presented. JNK-P is also key as presented by the authors.

5) The pharmacological experiments require some level of meaningful rescue as well: epothilone B rescuing JNK-P.

Important points that should be addressed:

1) RNAi experiments The authors present no phenotypes after 18 DIV using RNAi, but then at 26DIV -- again only Syt punctae as read-out – what is the overall morphology of these neurons. It is unclear what is happening to these cells from pictures. ato-RNAi has similar problems – in both cases it is unclear when neurons are actually running out of protein, and it is unclear whether there is a developmental defect or no rescue… These 'delay experiment' does not allow to conclude 'not only required for development, but also maintenance'.

2) kinesin and drug experiments. The kinesin (unc104) overexpression rescue of the shot, tau double mutant is a strong experiment. The authors show rescue of syt puncta at 2 DIV. As above, we are not sure what the overall morphology and issues of these cells are. Are these really fully rescued neurons? Unfortunately there is no in vivo experiment in this case. How about brp? Does it also rescue the 'aging/maintenance' issues?

The authors find that the microtubule-stabilizing drug epothilone B also provides a significant rescue of Syt 'intensity' in cultured neurons at DIV 2. Like the puncta graphs, everything is normalized to 100 in the graph (6E), but apparently now intensity and not numbers of puncta are compared? Why? Next the authors produce microtubule stress with nocodazole and report active JNK redistribution similar to the tau, shot double mutant and conclude that microtubule stress is the underlying cause. The key test of this idea would be epothilone B treatment to reduce microtubule stress and show that JNK-P is normalized, but unfortunately this experiment is missing.

3) JNK experiments

The attenuation of the JNK pathway itself as a partial rescue of the tau, shot double mutant is again a strong experiment. However, again the selective read-outs and different experimental approaches make it difficult to come to a straight-forward conclusion. For example, in the brain the authors use a DN for bsk to attenuate JNK signaling, but the same method is not used for comparison for the syt count experiments in DIV 2 neurons. Why is n-syb used as a read-out at the end of the paper for the first time? Furthermore, there is no information how either of the JNK attenuation approaches affects Unc104 localization and function.

---

## [Author Response]

*Essential revisions:*

The abstract and introduction might be toned down a little and move away from the idea of 'homeostasis'. This term has become all things to all folks, and 'synaptic maintenance' or similar phrasing could be used.

We toned down the abstract and introduction and exchanged the term 'homeostasis' for 'synaptic maintenance' throughout the manuscript

*The authors should take time within the results to explain the experimental system – i.e. the tau mutants are lethal, and the primary culture allows examination of neurons beyond this lethal stage.*

We have added the suggested changes in the text.

For Figure 1, we would suggest including, or referring to the later figure where double heterozygotes of shot/tau give significant data – this more directly alludes to the pathway, rather than two loss-of-function mutants making a defective phenotype worse (But see below on this). They also refer often to 'synapses' when a more precise description of (something like) 'materially deficient' synapses would be useful.

Thanks for pointing this out. When discussing the issue of redundancy, we now refer to Figure 5 and indicate that double-heterozygotes give significant reduction in synapses supporting the idea that tau and shot function in the same pathway. As further proof of redundancy, we are now including cross-rescue experiments between tau and shot (Figure 1—figure supplement 3).

*The most widely used method to count presynaptic sites in Drosophila is to stain them with specific antibodies against presynaptic proteins. Accordingly, a reduction in the number of Syt- or Brp-stained puncta is commonly interpreted as reduction in the number of presynaptic sites. We used this strategy with both synaptic markers in the first part of the manuscript in order to characterise the phenotypes. For the rest of the paper we mostly focused on Syt which has been used previously to quantify presynaptic sites (see as an example Wan et al., Neuron, 2000; Liu et al., The Journal of Neuroscience 2010; Franciscovich et al., Genetics 2008 and Rieckhof et al., Journal of Biological Chemistry, 2003). We have introduced a description of the phenotype, which is more precise: “[…]tau-deficient primary neurons contain fewer Brp and Syt positive presynaptic specialisations. In the following, we will refer to this phenotype as “synapse reduction”. We find that exchanging the term synapses by materially deficient synapses across the text would make the reading of the paper unnecessary complicated.*

The strongest experiments in the paper are the rescue of tau, shot mutants through unc104 overexpression and through JNK attenuation. However, in these cases, only selective read-outs are used (for example, we do not learn whether JNK attenuation affects unc104, as predicted by the model).

We performed new experiments both in culture and in embryos for the rescue of tau-shot mutants through JNK attenuation and stained for unc104. As the model predicts, we find that the aberrant localisation of unc104 in the somas is rescued upon JNK attenuation, and we demonstrate this both in vivo and in vitro. These data are now discussed in the results part and shown in Figure 8—figure supplement 1 and Figure 8—figure supplement 2.

*Sometimes read-outs focus on syt punctae, sometimes brp, n-syb and different ages in culture or* in vivo*. Different read-outs in different experiments make comparisons more difficult.*

At the beginning of the results part we look at both Syt and Brp punctae in the single and double *tau-shot* mutant neurons to demonstrate that several presynaptic proteins are affected. Subsequently, we consistently focus on Syt and, in some instances n-Syb is used for technical reasons (see below). We have made sure that in the text, the use of readouts and stages are clearly indicated.

*A principal concern arising from the presentation of selective read-outs is the specificity of the phenotypes. Defective syt puncta (the most commonly used read-out in the paper) can arise directly and indirectly from numerous causes. Just because they become more when there is more kinesin does not mean that the function of tau and shot is truly rescued.*

We provide many data that point to the involvement of kinesin-3 and defective transport in this synaptic function. These indications include the defects in transport, the defective localisation of Unc-104 in *shot-tau* mutant neurons, the genetic interactions between *tau*, *shot* and *unc-104* and the Unc-104 rescue experiments. These experiments support our model that diminished function of Unc-104 in axons is the cause for the synaptic defects and that synaptic defects are not the consequence of general ill health of neurons. We can also rule out that defective Syt puncta is an indirect consequence of defects in axonal length and number of branches of cultured neurons since our new analyses of morphological parameters show no correlation between synaptic phenotypes and change in morphology (see also the new passage in the Discussion).

As for the ability of kinesin-3 to truly rescue the function of Tau and Shot, we only claim that kinesin-3 rescues the synaptic phenotypes in double mutant neurons. Tau and Shot are microtubule regulators and regulate other aspects of neuronal physiology and morphology, which are independent of kinesin-3.

I am wondering what these neurons look like, what other markers look like and what aspects of neuronal function and maintenance are truly 'rescued' with unc104 overexpression compared to the microtubule-stabilizing drug compared to JNK signaling attenuation.

These data are now provided under "Required experiments" below.

*Finally, a key issue is the lack of simple genetic rescue experiments. The authors only provide one tau rescue experiments at the beginning of the paper. The shot mutant is never rescued by a shot transgene.*

These data are now provided, see next under "Required experiments".

*In sum, while we find the paper compelling and interesting substantial revisions are required and should be done within the two month period required by eLife.*

Required experiments:

1) Rescue of shot, rescue attempt of tau with shot and rescue attempt of shot with tau. Only these experiment reveal functional overlap.

We now include a series of rescue experiments. We found a very good rescue of the synaptic phenotypes with *shot* rescue in *tau* mutants, *shot* rescue in *shot* mutants and *tau* rescue in *shot* mutants, further confirming our hypothesis of functional overlap between these proteins. These results are now incorporated in the text and the data is shown in Figure 1—figure supplement 2.

*We do not think the rescue of the double mutant with either tau or shot is as meaningful, because the double mutant clearly gets worse than any single and any single rescue in that background would make it better, without revealing functional overlap of the two proteins.*

*2) Morphology of the neurons throughout, in mutants and rescues, to show to what extent syt puncta counts are a consequence of more severe neuronal defects.*

We incorporated the analysis of axonal length and number of branches of cultured neurons. We find that in the double mutant condition, the length of axons is not significantly affected but the number of branches is reduced. We performed similar analysis for the tau-shot RNAi experiments at 3, 18 and 26 DVI. In none of the time points we observed changes in axonal length nor in branch number of tau-shot RNAi neurons when compared to control neurons. This data clearly indicates that the synaptic phenotype occurs independently of the branching phenotype.

We further quantified axonal length and branch number in the rescues of *tau-shot* with unc104 overexpression and JNK attenuation. We find that axonal length is slightly reduced in both conditions and the number of branches remains significantly reduced in the unc104 rescues but is rescued upon JNK attenuation. Therefore, there is clearly no correlation between morphological changes and changes in synapse number. These data are explained in the results and presented in Figure 3—figure supplement 3 and Figure 3—figure supplement 4.

*3) Further regarding the specificity issue: Please see the reference in the review below to how the original unc104 paper made its point on specificity (mitochondria, HRP, fasII, mCD8, – subset of these). The key is whether this is a defect specific to synaptic proteins as proposed or a general transport or even health problem of these neurons.*

Since we can successfully rescue the synaptic phenotype at the various levels of the pathological pathway (and find no correlation with morphological changes affected differently in these rescues; see previous point), we are confident that the synaptic defects we observe are dependent on and specific for the pathway we describe and not the indirect consequence of general cellular decay. As discussed above, Tau and Shot are microtubule regulators and, as such, regulate other aspects of neuronal physiology and morphology. Accordingly, we do not claim at any point that Tau and Shot function exclusively regulates the transport of synaptic proteins.

*4) The authors should use UNC-104 as a readout in the rescues and throughout for mutants presented. JNK-P is also key as presented by the authors.*

We did new rescue experiments and tested the localisation of unc104. Both in vivo and in vitro, the aberrant localisation of unc104 in the somas is rescued upon JNK attenuation, as is consistent with our model. These data are now discussed in the results part and presented in Figure 8 S1 and S2. As for JNK-P we also confirmed that epothilone B rescues the localisation of JNK-Pin *tau-shot* i.e. it increases the localisation of JNK-P at axonal tips and reduces the aberrant localisation is somas (Figure 6—figure supplement 1).

*5) The pharmacological experiments require some level of meaningful rescue as well: epothilone B rescuing JNK-P.*

This experiment has been performed successfully.

Important points that should be addressed:

*1) RNAi experiments The authors present no phenotypes after 18 DIV using RNAi, but then at 26DIV -- again only Syt punctae as read-out – what is the overall morphology of these neurons. It is unclear what is happening to these cells from pictures. ato-RNAi has similar problems – in both cases it is unclear when neurons are actually running out of protein, and it is unclear whether there is a developmental defect or no rescue… These 'delay experiment' does not allow to conclude 'not only required for development, but also maintenance'.*

Detailed morphological data for RNAi experiments in cell culture have been added. In summary, at any of the time points: 3, 18 and 26 DVI we observed changes in axonal length nor in branch number of tau-shot RNAi neurons. For the ato-RNAi we use independent readouts to assess the number of axonal branches and find that precocious synapse decay is not due to axonal loss or to changes in branch patterns (Figure 3).

The reviewer wonders when neurons are actually running out of protein.To clarify this, we performed experiments in primary neuronal cultures with RNAi for *tau* and *shot* and stained with anti-Shot and anti-Tau at 3 DIV and 25 DVI. We find that at 3 DIV the intensity of anti-Shot is unchanged when compared to controls. By 25 DIV the intensity of anti-Shot is down to at least 50% when considering background levels in staining that are suggested by parallel controls with null mutant neurons. As for Tau, the intensity of anti-Tau is lower than controls at 3 DIV and it continues to decrease significantly between 3 and 25 days. These results support that this strategy takes out Tau and Shot functions with some delay, overcoming developmental functions (i.e. pre-synaptic sites can be properly established as well as axonal branches). These data can be found in Figure 3—figure supplement 1.

*2) kinesin and drug experiments. The kinesin (unc104) overexpression rescue of the shot, tau double mutant is a strong experiment. The authors show rescue of syt puncta at 2 DIV. As above, we are not sure what the overall morphology and issues of these cells are. Are these really fully rescued neurons? Unfortunately there is no* in vivo *experiment in this case. How about brp? Does it also rescue the 'aging/maintenance' issues?*

The outcome of our morphology studies of Unc104 rescue was described above and suggests that Unc104 only fully rescues the transport-related, synaptic phenotypes of the *tau-shot* double mutant neurons (now shown in Figure 3—figure supplement 4). The involvement of Unc104 is supported by rescue experiments but is further supported by transport assays, genetic interaction between *tau-shot* and *unc104* and the substantial changes in the localisation of Unc104 (as described already in the first manuscript version).

As for the aging/maintenance issue, we now also provide unc104 brain rescues of aging/maintenance phenotypes induced by tau-shot RNAi in vivo (Figure 5—figure supplement 2).

Concerning BRP, it has been previously demonstrated that Syt and BRP are both transported by unc-104, therefore it is expected that defects in Unc104 function will affect both synaptic proteins (Pack-Chung et al., 2007). To repeat the unc-104 rescues using BRP will take more than 2 months since the fly stocks need to be re-established. We feel that the advance provided by such data is not sufficient to justify the time investment.

*The authors find that the microtubule-stabilizing drug epothilone B also provides a significant rescue of Syt 'intensity' in cultured neurons at DIV 2. Like the puncta graphs, everything is normalized to 100 in the graph (6E), but apparently now intensity and not numbers of puncta are compared? Why?*

The experiments with epothilone B are not performed in cultured neurons but in embryonic motoraxons. The terminal of these axons at this developmental stage is very narrow and small; it is not possible to quantify separated puncta. To overcome this caveat, we quantified the intensity of synaptotagmin staining as we did similarly in Figure 2.

*Next the authors produce microtubule stress with nocodazole and report active JNK redistribution similar to the tau, shot double mutant and conclude that microtubule stress is the underlying cause. The key test of this idea would be epothilone B treatment to reduce microtubule stress and show that JNK-P is normalized, but unfortunately this experiment is missing.*

We now show that epothilone B treatment normalises JNK-P, these data are presented in Figure 6—figure supplement 1.

*3) JNK experiments*

The attenuation of the JNK pathway itself as a partial rescue of the tau, shot double mutant is again a strong experiment. However, again the selective read-outs and different experimental approaches make it difficult to come to a straight-forward conclusion. For example, in the brain the authors use a DN for bsk to attenuate JNK signaling, but the same method is not used for comparison for the syt count experiments in DIV 2 neurons. Why is n-syb used as a read-out at the end of the paper for the first time?

The JNK pathway can be attenuated in different ways, for example using mutations in *wnd*, expressing UAS-puc and Uas-bsk-DN. All these methods have been frequently used by others and are widely accepted. Our choice of method was for technical reasons. For culture studies, the *wnd* mutation and UAS-puc could be easier combined with the *tau-shot* mutant chromosome. For in vivo studies in the adult brain, the X-chromosomal Uas-bskDN was easier to combine with the existing Uas-tau-RNAi, Uas-shot-RNAi, ato-Gal4 and Uas-nSyb fly lines. Similarly, for technical reasons, we used nSyb instead of Syt in a few experiments in the adult brain. The reduction in nSyb we observed can be taken as a further confirmation for the phenotype described for Syt and is in agreement with previous reports demonstrating that nSyb is also transported by Unc-104 (Pack-Chung et al., 2007).

*Furthermore, there is no information how either of the JNK attenuation approaches affects Unc104 localization and function.*

We performed new experiments showing that JNK attenuation restores Unc104 to normal distribution patterns. We present these data in Figure 8—figure supplement 1 and Figure 8—figure supplement 2.